# Performance Assessment of Portable Optical Particle Spectrometer (POPS)

**DOI:** 10.3390/s20216294

**Published:** 2020-11-05

**Authors:** Fan Mei, Gavin McMeeking, Mikhail Pekour, Ru-Shan Gao, Gourihar Kulkarni, Swarup China, Hagen Telg, Darielle Dexheimer, Jason Tomlinson, Beat Schmid

**Affiliations:** 1Pacific Northwest National Laboratory, Richland, WA 99352, USA; Gourihar.Kulkarni@pnnl.gov (G.K.); swarup.china@pnnl.gov (S.C.); jason.tomlinson@pnnl.gov (J.T.); Beat.Schmid@pnnl.gov (B.S.); 2Handix Scientific LLC, Boulder, CO 80301, USA; gavin@handixscientific.com; 3NOAA Earth System Research Laboratory, Chemical Sciences Division, Boulder, CO 80305, USA; RuShan.Gao@noaa.gov; 4Cooperative Institute for Research in Environmental Sciences (CIRES), University of Colorado, Boulder, CO 80309, USA; hagen.telg@noaa.gov; 5Sandia National Laboratories, Albuquerque, NM 87185, USA; ddexhei@sandia.gov

**Keywords:** optical particle spectrometer, aerosol size distribution, instrument characterization

## Abstract

Accurate representation of atmospheric aerosol properties is a long-standing problem in atmospheric research. Modern pilotless aerial systems provide a new platform for atmospheric in situ measurement. However, small airborne platforms require miniaturized instrumentation due to apparent size, power, and weight limitations. A Portable Optical Particle Spectrometer (POPS) is an emerged instrument to measure ambient aerosol size distribution with high time and size resolution, designed for deployment on a small unmanned aerial system (UAS) or tethered balloon system (TBS) platforms. This study evaluates the performance of a POPS with an upgraded laser heater and additional temperature sensors in the aerosol pathway. POPS maintains its performance under different environmental conditions as long as the laser temperature remains above 25 °C and the aerosol flow temperature inside the optical chamber is 15 °C higher than the ambient temperature. The comparison between POPS and an Ultra-High Sensitivity Aerosol Spectrometer (UHSAS) suggests that the coincidence error is less than 25% when the number concentration is less than 4000 cm^−3^. The size distributions measured by both of them remained unaffected up to 15,000 cm^−3^. While both instruments’ sizing accuracy is affected by the aerosol chemical composition and morphology, the influence is more profound on the POPS.

## 1. Introduction

### 1.1. Summary of Optical Particle Spectrometer Application

Atmospheric aerosols have a broad size range, which covers more than four orders of magnitude. From newly formed nanometer clusters to super-micron cloud droplets and dust particles, ambient aerosols actively affect and are affected by atmospheric processes and human activities [1,2,3]. High-quality measurements with suitable temporal and spatial resolutions are essential for addressing the research questions in meteorology, atmospheric processing, severe weather monitoring, and in many other areas of human activity, like industrial hygiene and the semiconductor and pharmaceutical industries [4,5,6,7]. A high-resolution real-time technique is critical to capture aerosol properties and their space and/or time variations, especially from a fast-moving platform (manned or unmanned). Optical Particle Spectrometry (OPS)—a technique based on elastic light scattering—has served as a foundation for such a real-time measurement in a compact form for a few decades [8,9,10].

Various OPS based sensors have been developed, and their performance, errors, and uncertainties were evaluated [11,12,13]. Several commercial OPS instruments emerged to provide a portable personal monitoring solution in the semiconductor and pharmaceutical industries [14]. Another application of those OPS includes collecting meaningful scientific measurements from atmospheric field studies, such as the deployment of a Passive Cavity Aerosol Spectrometer (PCASP, DMT, CO, USA) and the Ultra-High Sensitivity Aerosol Spectrometer (UHSAS, DMT, CO, USA) [11,15,16,17,18].

Meanwhile, the rapid growth of the unmanned aerial system (UAS) platforms and applications has also expanded the demand for OPS deployment in atmospheric research to provide high-resolution temporal and spatial datasets [4,5,6,7]. While many commercial compact instruments are available on the market, most of those sensors are not suitable for UAS-borne atmospheric research due to size, weight, data acquisition, and power requirements. Many researchers have focused on miniaturizing aerosol, gas phase, and cloud instruments for UAS applications in this vein [19,20].

A miniaturized and high sensitivity OPS—Printed Optical Particle Spectrometer (POPS)—has been developed by the National Oceanic and Atmospheric Administration (NOAA); it is manufactured using a 3D printing technique (which is reflected in the original name for the instrument). Several copies of POPS have been initially built and tested in the laboratory at NOAA [21]. The commercial version of POPS, renamed to the Portable Optical Particle Spectrometer by the vendor (Handix Scientific), comes in several packages depending on the intended application. The light-weight particle profiler (LW2P) model includes a light-weight enclosure and hardware for an autonomous airborne operation intended for use with UAS. The compact optical particle spectrometer (COPS) model features a hardened weatherproof enclosure for a tethered balloon system (TBS) deployment.

However, the long-term instrument performance and its sensitivity to the operating environment are still unknown. Further tests in a controlled environment are necessary to provide guidance on this instrument’s operation and deployment on a UAS or TBS platform. This paper described evaluating the variability of operational parameters using the commercial versions of POPS, especially with the laser heater and temperature sensor upgrade. While we didn’t directly test the counting efficiency under extremely low temperatures or pressures due to the facility limitation, the current lab evaluation and the successful deployment of the POPS with the specially designed Arctic enclosure provided us the confidence in the POPS performance. Implications of aerosol refractive index on size distributions provided by POPS are discussed. Application of the POPS in real atmospheric research is illustrated by a case study from a TBS deployment at the South Great Plain (SGP) observatory of the Department of Energy (DOE) Atmospheric Radiation Measurement (ARM) user facility.

### 1.2. Portable Optical Particle Spectrometer (POPS)

Rapid changes in the Arctic have prompted the DOE ARM Aerial Facility (AAF) to expand the scope of climate research activities using a tethered balloon system (TBS) or unmanned aerial system (UAS) on Alaska’s North Slope. Several POPS (Handix Scientific) of different versions have been procured to serve as routine instruments for the AAF UAS and TBS deployment. Its design has evolved and differs in minor ways from the original system described in Gao et al., 2016 [21]. Changes include modification to the inlet/exhaust mechanical design, implementation of round versus square optical elements, and improvements to circuit board design, software and firmware. More recent changes include replacing several 3D printed components used for mounting the laser and optical elements with machined aluminum parts, which was implemented to reduce contamination of the optical surfaces by apparent outgassing of plastic material.

A 405 nm laser diode is used in the POPS as a light source. A laser heater has been added to the commercial (Handix) version, as well as to later NOAA-produced systems, to maintain the laser operating temperature at a minimum of 35 °C, which becomes essential in cold weather applications. The NOAA statistical analysis of the uncertainty on hypothetical aerosol populations indicated that the errors are less than 1.5% for the accumulation mode parameters (peak height, location, width) and number concentration for the aerosol sizes between 140 and 3000 nm [21]. Both Handix versions feature identical instrument designs and measure the optical size of sampled particles using single-particle light scattering in the range of 140 nm to 3 μm.

The sampling flow is sheathed in clean air. It enters the optical chamber to maintain a laminar state and prevent particles from entering the fringe area of the sampling volume, where sizing becomes biased. Nominal (on the ground) sampling and sheath flows are 3 cm^3^/s and 9 cm^3^/s, respectively. When operated on an aircraft or UAS platform, where the inlet lines’ pressure may differ from the cabin and/or ambient pressure, the POPS sample and sheath inlets are connected to the same inlet manifold to minimize the flow distortions. The exhaust line also needs to be balanced with a similar pressure as the inlet line to reduce the miniature rotary vane pump load.

An additional temperature sensor is installed after the optical chamber’s exit port to monitor the operational temperature, as discussed in Section 3. When deploying to the icy environment, an insulated enclosure was designed to keep the POPS internal temperature around 15 °C.

The POPS reports several parameters useful for evaluating the overall instrument performance. The most important is the standard deviation of the instrument baseline signal, which is essentially the instrument’s noise level. The instrument baseline is the photodetector signal measured in the absence of particles. It depends on several factors, including Rayleigh scattering, electronic noise, and light scattered off interior surfaces in the instrument [21]. The noise level and its fluctuations constrain the smallest particle diameter that can be measured with confidence. The initial instrument baseline and standard deviation are measured by the manufacturer for filtered, particle-free air prior to calibration. The initial baseline value should not be modified unless a significant baseline drifting is observed. The baseline value set the trigger point for recording valid particle pulse data. Reducing the value will lead to a higher data recording rate and potentially false particle counting. Increasing the values will prevent the system from acquiring real particle data measured at the lower range of the instrument detection window.

## 2. Materials and Methods

The characterization of aerosol spectrometer instruments has two aspects: Particle sizing accuracy and particle counting efficiency. The counting efficiency characterization is typically performed via comparison with a reference instrument since there are no absolute standards for aerosol particle concentration in the air. In this study, we used a Passive Cavity Aerosol Spectrometer Probe (PCASP) and condensation particle counter (CPC) model 3772 (TSI) as reference instruments. The PCASP is a full-size instrument primarily designed for airborne applications with intensive deployment history in atmospheric research [22]; it provides a size-resolved aerosol concentration range between 0.10 and 3 μm. An open cavity HeNe laser of 632.8 nm is used as a light source in the instrument.

The accuracy of particle sizing is usually characterized using a test aerosol of known size. For optical instruments with a well-defined light scattering cross-section, the size accuracy also depends on particle size and morphology, refractive index, and wavelength(s) of the tested instrument’s light source. One of the commonly used calibration aerosols is polystyrene latex (PSL) spheres, the physical parameters of which are well known and commercially available in many but limited number of sizes. Another conventional way to produce a specific size test aerosol is to extract particles of the desired size from aerosol of wide size distribution using a size selecting technique, for example, electrical mobility sizing. In this study, we have used PSL and ammonium sulfate (AS) particles aerosolized via atomization from water suspension or solution, as well as Arizona test dust (ATD), silicon dioxide nanopowder (spherical, porous), and titanium dioxide particles aerosolized from dry powder (see Table 1); all test aerosol was size selected with the help of Differential Mobility Analyzer (DMA) column (TSI, model 3081).

It should be noted that optical sizing instruments provide not a real physical (“geometric” or “equivalent volume”) size but rather an “equivalent optical size,” one of the definitions of which is “diameter of a calibration particle that scatters as much light in a specific instrument as the particle being measured” [30]. Equivalent optical size may substantially deviate from the geometric one if the particle refractive index is different from the refractive index of aerosol used for calibration (in our case PSL) and/or if the particle shape deviates too much from a sphere. Equivalent volume diameter could be estimated from equivalent optical diameter using theoretical response function or calculations (e.g., Mie or T-matrix method) for specific sensor optical geometry, assumed refractive index, and particles’ morphology [31,32].

Scanning Mobility Particle Spectrometer (SMPS) differentiates aerosols according to equivalent electrical mobility size, which is close to equivalent volume size for most common aerosols of near-spherical shape.

For adequate comparison of aerosol sizes obtained by different instruments, the sizes should be converted to the one standard type, for example, to volume equivalent diameter (“the diameter of a sphere having the same volume as the irregular particle” [33]).

The typical relationship between equivalent volume *D_v_* and electrical mobility *D_m_* diameters is (see e.g., [34]):(1)$$D_{v}=D_{m}\frac{C_{c}\left(D_{v}\right)}{C_{c}\left(D_{m}\right)}\frac{1}{\chi_{t}}$$
where *C_c_*(*D*) is Cunningham slip correction factor for diameter *D*, $\chi_{t}$ is a dynamic shape factor which “relates the motion of a particle under consideration to that of a spherical particle” [30].

The aerosol particles measured by each instrument are usually presented as distribution among the various sizes. In this study, we used the geometric mean diameter (*D_pg_*) to describe the tested aerosol’s sizing property.
(2)$$D_{pg}=exp\left(\frac{\sum n_{i}\times lnD_{pi}}{N}\right)$$
where *n_i_* is the number of particles in size bin *i*, having a midpoint of size *D_pi_*, and where $N=\sum n_{i}$.

Several studies have been devoted to evaluating the performance of a DMA with aerosols of irregular shape [35]. It has been shown that for dry dispersed mineral dust aerosols (like ATD), the size selected output of a DMA has broader than expected distribution, smaller mode size, and contained a significant amount of particles with sizes beyond expected passband [35].

The schematic drawing of the apparatus for all tests carried out in this manuscript is shown in Figure 1. The diagram includes three major sections: The aerosol generation, the primary aerosol selection, and the characterization section (two different setups). Aerosols generated from atomizer (model 3076, TSI) or dust aerosol generator (SSPD 3433, TSI) passed through a dilution system to limit total concentration within the instruments’ operation limits. They then passed through the first DMA (model 3081, TSI) to produce monodisperse test aerosol particles. In the characterization setup 1, the sheath flow to aerosol flow ratio in the first DMA was kept 6:1 for 500 nm particles and 10:1 for particles less than 300 nm. In the characterization setup 2, the sheath flow to aerosol flow ratio in the first DMA was kept 10:1, and 5:1 for the 2nd DMA for all tests. The size-selected aerosol was simultaneously measured by POPS and a reference sensor, CPC or PCASP in characterization setup 1. In setup 2, the reference sensors were UHSAS (Ultra-High Sensitivity Aerosol Spectrometer, DMT) and an SMPS (Scanning Mobility Particle Spectrometer, model 3938, TSI, 2nd DMA combined with CPC 3775, TSI). During the relative humidity tests and dust particle tests, a 30 L aerosol conditioning chamber was attached before three instruments, as shown in the schematic (Figure 1). The chamber temperature was kept at room temperature. This conditioning chamber provides sufficient time for the aerosol to gain equilibrium and uniform mixing within the sampling airflow.

The manufacturer calibration procedure follows a similar principle but uses different instruments for conditioning test aerosol and as a reference sensor. The specific operation parameters are included in the supplemental document with the typical counting efficiency plot in Appendix A.

## 3. Results and Discussion

### 3.1. Counting Efficiency at the Lower Size Range

The counting efficiency is defined as the ratio of the particle number concentration counted by POPS to the particles’ total particle number concentration within a specific size range. We have used CPC (model 3772, TSI) as a reference sensor for counting efficiency tests to measure the total number concentration. Figure 2 shows the counting efficiency measured at “ground” pressure (~1030 hPa) with ammonium sulfate particles in the range of 135 nm to 300 nm (electrical mobility diameter). The *y*-axis error bars indicate the standard deviation of the counting efficiency values. We did not put *x*-axis error bars in the figure. Still, they are theoretically equal to the sizing uncertainty of the DMA (±2%), which was estimated from the square too of the sum of squares of the sheath flow, pressure, temperature, and voltage [36,37].

The total number concentration of aerosol particles is more sensitive to the counting efficiency change between 135 and 300 nm. The lower size range (between 135 and 300 nm) aerosol particles usually significantly contribute to the atmospheric aerosol population. While we were unable to test the counting efficiency of the ammonium sulfate particles in the full POPS size range due to the DMA sizing limitation, the counting efficiency of this POPS held close to 100% when we tested with 707 ± 9 nm, 1019 ± 15 nm, and 2020 ± 15 nm PSL particles. The decrease in the counting efficiency curve at the lower end of the detection range reflects diminishing probability to detect a weak light pulse generated by a small particle over the instrument background noise level. Variations in light intensity across the sensing volume, instability of the light source, and deviation of the particle trajectory from the beam axis also affect the strength of a scattered signal, which may render it below the detection limit. We have found that the counting efficiency curve shown in Figure 2 holds when the difference between the ambient pressure and the inlet pressure is less than 150 hPa. When the pressure difference exceeds 150 hPa, we observed a size distribution distortion, which is likely due to the printed optical chamber deformation. Replacing the printed optical chamber with metal material can prevent this issue, and such modified POPS has been implemented and deployed by other researchers [38].

The lower detection limit or threshold diameter is usually defined as the particle diameter, where 50% of the asymptotic maximum counting efficiency is reached (common designation is *D_50_*). The current study demonstrated an example on the data obtained from one (SN009 from the early production) of four units of POPS (SN009, SN014, SN018, and SN139) using the size selected AS particles and PSL particles for characterization and D_50_ = 135 nm with the ammonium sulfate particle, The statistics from 22 instruments manufactured in 2020 shows the average value of D_50_ = 129 ± 6 nm using PSL particles for the characterization.

The average standard deviation of the baseline measured for the above 22 POPS instruments was 9.3 ± 2.1 digitized counts (noise in the raw signal), while the average baseline value was 2170 ± 110 digitized counts. Theoretically, these values guided the determination of the minimum detectable raw particle peak signal, which corresponds to a minimum PSL diameter. However, to minimize the occurrence of the false-positive particle detection, the effective particle sizes (*D_50_* values) usually exceed noise limits due to the artifacts related to laser beam profiles and vary from instrument to instrument. An example of a manufacturer efficiency curve provided by the vendor is shown in Appendix A. The inlet and exhaust ports of the instrument under test have been modified, as described in Section 1.2. Still, the instrument shares the basic design and functionality of other PNNL units and NOAA prototypes.

### 3.2. Total Concentration Counting Limits

Concentration comparison was carried out using setup 2 (simultaneous sampling by POPS, SMPS and UHSAS) with the size-selected ammonium sulfate aerosol particles (200 nm) in the total concentration range of 20–15,000 cm^−3^. The results are summarized in Figure 3a and Appendix A. It should be noted that POPS usually has a noise peak below 135 nm, and this peak encompasses a total of about 170–200 cm^−3^ for the whole tested concentration range. To avoid these “ghost” counts, we set a lower limit on the POPS data to 135 nm for the rest of the comparison in this section, and further discussion on this issue is addressed later in more detail. For concentrations below 2000 cm^−3^, POPS and UHSAS measurements generally agree with the SMPS measurement (less than 10% variation). We observed about 25% of difference when the SMPS concentration reached 4000 cm^−3^, and the discrepancy between the SMPS and POPS/UHSAS enlarged with the increasing of the aerosol particle total concentration. This difference is mainly due to the counting coincidence losses. However, we noticed the baseline standard deviation remained the same up to 6000 cm^−3^, and the baseline still contained in the range of 2170 ± 110 digitized counts. Since both UHSAS and POPS experienced a similar trend of the total concentration undercounting, the differences between UHSAS and POPS remained within ±10% from each other through the entire concentration test range.

Comparison of the aerosol size distributions at two total concentrations (around 700 cm^−3^ and 15,000 cm^−3^) measured by POPS and UHSAS did not show significant size-shifting, as shown in Figure 3b and the geometric mean diameters calculated for the above two cases. The secondary peaks (>250 nm) in the POPS and UHSAS size distribution are due to the multiple charge effect on the DMA selected particles and the usual artifact of the electrical mobility-based separation systems. While both UHSAS and POPS experience a significant undercounting issue at 15,000 cm^−3^, the size distributions measured by both of them remained unaffected.

### 3.3. Size Determination

Two aerosol test particles (ammonium sulfate and PSL particles) were used to measure POPS responding signal. The theoretical optical signals calculated using Mie theory are compared with POPS measured response curve in Figure 4. The Mie calculations were performed using appropriate indices of refractions (see Table 1) at the assumed laser wavelength of 405 nm with proper account for scattering angles according to the internal optical chamber geometry and were scaled to the measured values. Similar to the calibration results from dioctyl sebacate (DOS) [21], oscillations in the scattering signals with particle diameter above 700 nm, caused by Mie resonances, can contribute about 30% to uncertainty in scattering amplitude. To simplify comparison algorithms, we used a “look-up-table” to establish a one-to-one relationship between the scattering amplitude and particle size at the size range >700 nm. For particle sizes <700 nm, we will use spline interpretation between the AS or PSL calibration data points to determine a one-to-one relationship between the scattering amplitude and particle size. The size distributions in the rest sections are all derived based on the above approach. The corresponding curve of two POPS with the PSL particles is included in Appendix A.

Using the characterization setup 2 (Figure 1), we also compared POPS, UHSAS and SMPS responses for the size-selected ammonium sulfate aerosol particles. To avoid repetition, we presented only 500 nm results in this manuscript. Note that a newly manufactured POPS (SN139) was used for this testing. The two instruments (the POSP and the UHSAS) observed similar size distributions of ammonium sulfate aerosols consistent with the expected output of the DMA, which provides validation to the correctness of calibration of both optical instruments (Figure 5).

The widening of the SMPS size distribution is due to the low sheath flow to the aerosol flow rate ratio (5:1) used for this case. The resolution of a DMA is described as the ratio of the electrical mobility at the peak of the column transfer function to the full width of the transfer function at 1/2 of its maximum value, which depends on the aerosol flow and sheath flow ratio for nondiffusive particles [39]. The left “shoulder” peak in the SMPS size distribution (300–400 nm) was mainly caused by missing multiple charge correction for the full-size range because the maximum size range of the SMPS scan ends around 750 nm. The UHSAS is based on a laser with a 1054 nm wavelength, which leads to a monotonically increased theoretical response curve for the size range between 60 and 1000 nm divided into 99 size bins [15]. The POPS data were obtained with 32 size bins in these tests. Both determined sizes (based on the equivalent optical diameter) by POPS and UHSAS are smaller than the SMPS size (based on the electrical mobility diameter). A secondary peak (the D_pg_ was recorded in the parenthesis in the legend in Figure 5) at the larger sizes on the UHSAS size distributions is the usual artifact of electrical mobility based separation systems; it is formed by the multiple charged particles passed through the DMA column [36]. Substituting the calculated geometric mean diameter from Equation (2) into Equation (1), we can correlate the average equivalent optical diameter and the average electrical mobility diameter, assuming the dynamic shape factor of ammonium sulfate particles is 1.07 [40]. Additionally, we noticed a 5 ± 2% sizing difference between the POPS and UHSAS, which most likely due to the error propagation with the size determination algorithm using the Mie calculation. Note that the increase in the number of the POPS size bins may cause the sizing issue due to the POPS complex size determination, as shown in Appendix A.

### 3.4. Particle Morphology and Refractive Index Effects on POPS And UHSAS

In this study, we used two test aerosols with a wide variety of refractive indices and morphologies (see Table 1); the sensors from setup 2 were used. Both SMPS and UHSAS were the reference sensors. The test aerosol of 500 nm was created and conditioned using the same apparatus for counting efficiency study (Figure 1). The same newly manufactured POPS (SN139) was used for this testing. It should be noted that the chosen diameter falls between wavelengths of the sensors’ lasers: 405 nm and 1054 nm. This diameter of 500 nm is far enough from the roll off of POPS’s counting efficiency curve at the lower end, but still well below the area where Mie theory predicts oscillations of the scattered signal for most common aerosols (e.g., aerosol particles with the refractive index below 2).

Morphology of the irregular shaped particles (ATD and TiO_2_) was examined with a scanning electron microscope; examples of particle images are presented in Appendix A.

Figure 6 and Figure 7 show examples of size distributions measured by POPS, SMPS, and UHSAS for 500 nm (electrical mobility diameter) test aerosols: ATD and TiO_2_. In all cases, POPS showed the presence of particles in the lower size range (less than 170 nm in this case). This artifact was previously observed by the manufacturer, especially on units with marginal laser profile quality. The artifact is probably caused by false positive detection associated with oscillations in the baseline signals near “true” particle events. The developer (NOAA) and the manufacturer (Handix Scientific) have implemented the particular procedure in the updated firmware to minimize the false detections, with more recently manufactured units displaying few, if any, false particle counts near the lower detection limit. However, despite these improvements, readings from the lowest bins should be interpreted with care; monitoring the baseline and its standard deviation is advised.

The tests with the two aerosols of irregular shape (Figure 6 and Figure 7) showed significant differences in POPS, SMPS, and UHSAS responses. The D_pg_ was calculated for all three size distributions. Again, we observed the secondary peak (the D_pg_ was recorded in the parenthesis in the legend in Figure 6 and Figure 7) at the larger sizes on the UHSAS size distributions. However, we couldn’t identify the clear primary peak in the POPS size distribution. For both ATD and TiO_2_ particles, the D_pg_ calculated from two modes from the UHSAS were smaller than the ammonium sulfate particles in Figure 5. Meanwhile, due to the “smeared” size distribution of POPS, the POPS D_pg_ values were larger than the SMPS D_pg_ values. Several factors could have contributed to these discrepancies. Differentiation of irregularly shaped aerosols with a DMA often does not produce particles of desired size (e.g., close equivalent volume diameter), yields broader size distributions than expected from the ratio of sample to sheath flow, and produce a significant number of particles larger then DMA bandwidth [35]. ATD is a mixture of several mineral species of different physical properties (notably refractive index, shape factor, and density); it is usually characterized by some “effective” set of parameters. For example, the value of the complex refractive index we used is 1.51 + 0.001i (Table 1); however, single particle in the mixture may have a refractive index from 1.413 + 0.000773i (illite) to 3.102 + 0.0925i (hematite) [26]. Particle dynamic shape factor for ATD may vary in a wide range from 1.3 to 3.1 [35], which directly affects particle size selected by DMA, since equivalent mobility diameter depends inversely on the dynamic shape factor, see Equation (1) [34]. All these processes contribute to the “smearing” of the measured size distributions, as visible in the POPS data.

UHSAS size distribution for the TiO2 test, shown in Figure 7, illustrates the notable refractive index effect on the observed distributions. The UHSAS used in the study was calibrated with PSL particle; a significant shift of the size distribution to the left is accounted for by the relatively high value of the refractive index of TiO2 particles (2.682 compared to 1.63 for PSL). The theoretical signal response curve of UHSAS for different refractive indices was presented by Cai et al., 2008 [15] (Figure 1 therein). It should be noted that the theoretical response curve of UHSAS exhibits Mie oscillations at a rather large size range (above 3 um) due to the larger wavelength of the UHSAS laser; so the whole size distribution, including the portion of “doubly charged or triply charged” DMA artifacts, lays within “monotonous” part of the response curve. Therefore, the distribution is not distorted but just shifted down. In contrast, significant Mie oscillations appear as low as 700 nm on the POPS response curve (Figure 4) for PSL and should start at lower sizes (~200 nm) for aerosols with a higher refractive index (Appendix A). This uncertainty in sizing results in additional distortions of the POPS measured size distribution besides evident shift due to a higher refractive index. Note that the aerosol particle morphology and refractive index effects on the PCASP were more complicated than their influences on the POPS and UHSAS, as discussed in the supplemental document.

### 3.5. Laser Temperature Effect on POPS Operation

The diode laser temperature fluctuation affects the laser beam intensity, therefore causing errors in the scattering signal, which are not compensated since POPS has no independent sub-system to monitor and account for the laser power variation. The thermal deformation of the optical chamber would cause beam misalignment, which also contributes to changes in light intensity within measuring volume and possible “stray beam” reflections. A laser heater was installed into the current version of the POPS to stabilize the temperature of the laser and help maintain the temperature of the optical chamber.

This temperature effect on POPS performance was examined in the following experiment to mimic the laser failure situation in the field and guided the data quality control process. The POPS (SN 009) sampled 500 nm PSL particles with a laser heater deactivated while placed inside a temperature-controlled chamber at the temperature of 0 °C. No degradation of POPS performance was observed when the laser temperature was higher than 30 °C. After the laser temperature dropped below 25 °C, artifact signals appeared lower detection limits and significantly contributed to the total number concentration. Meanwhile, the primary PSL signal peak remained intact around 500 nm. With the laser temperature decreased below 20 °C, about 10% variation on the primary peak of PSL particles was observed. Thus, for the cases when laser temperature dropped below 25 °C, all the data for sizes below 150 nm should be discarded or flagged as “questionable.” As shown in Figure 8, with a decrease of the POPS laser temperature, the artifact signal appeared around 130 nm and grew both in concentration and size.

The artifact signal seems to be caused by the misalignment of the laser beam due to the thermal deformation of the printed optical chamber expected under extreme ambient conditions (winter, Arctic, or high altitude deployments). Thus, we recommend setting the laser heater temperature at least 30 °C to minimize the error in scattering signal and flag the data when the laser temperature deviates outside the normal working range of 25 °C–60 °C. This temperature setting is now the default setting on all AAF POPS instruments.

### 3.6. POPS Operation under Different RH Environment

Relative humidity (RH) may significantly affect aerosol physical and optical properties [41,42,43,44,45]. Because aerosol particles that contain hygroscopic material can absorb ambient water at elevated RH levels, as a result, ambient aerosol particles’ size, weight, and optical properties all vary as functions of RH. The POPS does not include a built-in drying module.

To evaluate the RH effect on the POPS performance, we subjected the dried PSL particles (500 nm) and 200 nm ammonium sulfate particles to variable RH conditions using the same residence chamber, as mentioned in Section 2. Water-vapor saturated airflow was created through a nafion humidifier, then circulated back to the residence chamber to change the residence chamber RH value. The test aerosol was fed into the residence chamber, where the level of relative humidity was controlled by varying the water-saturated airflow. Residence time in the chamber was long enough for the aerosol’s RH level to get fully equilibrated with the chamber RH condition. As expected, the change in RH did not significantly affect the PSL particles, as shown in Appendix A, because the polystyrene material is hydrophobic. We did not observe any degradation in POPS performance. Ammonium sulfate test aerosol showed the expected hysteresis behavior of deliquescence and efflorescence when relative humidity was allowed to change from 40% to 90% and back to 40%, as shown in Appendix A [46,47]. However, efflorescence transition was slightly masked by an influx of fresh (dry) particles. The maximum growth factor (the ratio of the aerosol peak size between the condition under the maximum testing RH and the condition of RH = 40%) for POPS and UHSAS (around 1.21) is smaller than the SMPS derived one (1.54), which is lower than the other growth factor study with the differential mobility technique. This artifact is due to the mismatch of the RH values between the sheath flow (<the chamber RH condition) and the aerosol flow (≈the chamber RH condition) in the SMPS system [46,48]. The smaller growth factor for the optical instruments could be explained by the change of the aerosol refractive index due to humidification (aerosol surface chemical composition will change due to the water absorption); the humidified particles were sized much smaller because they produced smaller scattering signal due to lower refractive index [49]. The above observation suggests that when comparing the POPS size distribution with other aerosol sizers, it is critical to examine them under the dry condition, commonly accepted below 40% in the RH level.

There are several ways to control the relative humidity of the aerosol samples. The most widely used approach takes advantage of the aerosol inlet flow temperature change via the sampling path. Using the August–Roche–Magnus approximation [50,51,52], we calculated the desired inlet temperature to examine if we can maintain the aerosol humidity below 40% (dry condition) when POPS operated at the Arctic or hot/humid conditions, like summertime SGP (Appendix A). The hygroscopic growth below 40% RH is limited and usually attributes less than a few percent of diameter change compared to the dry condition. This guideline can help us evaluate and interpret the field data, which the ARM user facility has provided worldwide.

For example, when TBS POPS were deployed at Oliktok Point, AK, the POPS inlet path and the optical chamber were packaged in an insulated enclosure and kept at a higher temperature to ensure the proper operation conditions were achieved. Based on one Arctic observation site data between April and October, the ambient temperature and RH at the ARM mobile facility (AMF, Oliktok Point, AK) is between −25 and 15 °C, and 20% and 95%, respectively. During the deployment at Oliktok Point, AK, the POPS optical chamber temperature was typically maintained at least 15 °C above the ambient temperature, ensuring the POPS measurements were under <40% RH. Between April to October 2018, the ambient temperature and RH at the ARM observatory at the SGP site ranged between 0 and 40 °C and between 20% and 90%, respectively. To make sure the measured aerosol particles were measured under the dry condition at the SGP site, the temperature difference of 20 °C (chamber temperature minus ambient temperature) was generally required. However, a high inlet temperature inside the enclosure and optical chamber will evaporate the volatile chemical species of the ambient aerosols. It is not recommended for the field operation if an external dryer system can be implemented.

With parallel inlet RH conditioners, several POPS could measure size distributions at various humidity conditions, which is of current interest to climate research. For example, a set of POPS with different inlet conditioners has been combined with other measurements to derive a comprehensive set of aerosol properties, such as aerosol optical depth or single scattering albedo, as described by Telg et al. 2017 [53]. However, if the parameters of dry aerosol are desired, an active humidity controller (dryer or heater) should be applied to reduce RH of the sample air.

## 4. Field Applications

### 4.1. POPS and PCASP Inter-Comparison During Aircraft Testing

POPS instruments have been compared to a variety of instruments in laboratory settings, e.g., scanning mobility particle spectrometer (SMPS, TSI) and aerodynamic particle sizer (APS, TSI) [21]. However, the airborne performance of POPS has not been tested in detail. As a part of this POPS evaluation, we deployed the instrument aboard a DOE research aircraft—the Gulfstream 159 (G-1)—for a flight on 17 August 2016. The G-1 was equipped with a comprehensive suite of meteorological, aerosol, trace gas, and cloud instruments [54], including the PCASP probe. The POPS was installed in the aircraft’s main cabin and sampled ambient aerosol via the isokinetic inlet. The PCASP was installed in its standard position, on the right-hand side wing pylon, and sampled through its standard passive inlet (expansion cone). Both instruments were calibrated using PSL particles before the flight. Locations of the probes and different sampling routes introduced additional uncertainty to this comparison—we did not account for a possible difference in particle loss rate in the inlets or a slight time delay that was introduced due to limited residence time in the sample supply lines for the POPS.

Figure 9 presented a comparison of the total number concentration and averaged size distribution obtained by both probes. The total number concentration was calculated by integrating the aerosol size distribution over an overlapping size range (between 0.14 and 3 μm). The time series and scatter plot (Figure 9a) demonstrate good agreement between POPS and PCASP. The correlation coefficient between 17:45 and 19:00 for the total number concentrations from two instruments is 0.9.

The size distributions were averaged over the period of 18:00:25 to 18:01:53 UTC when the G-1 was in a straight and level flight leg. The POPS had more counts than PCASP in the sizes below 200 nm. Several factors may contribute to this discrepancy: Different distortions to the aerosol size distribution due to differences in sampling conditions (isokinetic for POPS and pseudo-isokinetic for PCASP) or internal losses inside of POPS and PCASP.

### 4.2. TBS Deployment

One driver for the development of the POPS was the increasing need of the atmospheric research community for a miniaturized suite of aerosol instruments suitable for deployment on a small UAS or TBS along with other instrumentation (meteorological, radiation, etc.).

The TBS group from Sandia National Laboratories has developed a TBS system [55,56]. The current TBS aerosol payload includes three TBS impactor (TBI) packages, two Printed Optical Particle Spectrometers (size range of 0.14 to 3 µm), one condensation particle counter (model 3007, TSI, D_50_ = 10 nm), and a meteorological radiosonde package (iMet 4-RSB).

A TBI contains a four-stage impactor (Sioutas Personal Cascade Impactor, SKC) equipped with a programmable low-pressure drop pump and an aerosol inlet. Four stages of the impactor have the particle cut-off sizes: 2.5 µm, 1 µm, 0.50 µm, and 0.25 µm.

Two POPS are suspended on the tether line. One POPS is usually operated just below the balloon to reach the maximum possible altitude, which is ideally above the cloud top. A second POPS is generally operated lower on the tether, for example, near the cloud base. Both POPS are characterized before and after each deployment to ensure the consistency of the data. A CPC has usually been attached 0.5 m away from the POPS directly below the balloon.

Two TBI’s are usually attached to the tether about 30 cm away from each POPS. A third TBI is generally operated 3 m above the ground to provide a surface reference.

Overall, this TBS aerosol system provides the number concentration, the size distribution above 135 nm, and the aerosol samples for off-line analysis of size-resolved chemical composition.

The TBS team completed 17 aerosol-related flights at the SGP site in July and nine flights at the Oliktok Point site in August 2019. Three TBIs collected hundreds of substrates, and these samples were sent to the Environmental Molecular Sciences Laboratory (EMSL) facility for further off-line analysis. Samples were selected for study based on event-based meteorological data.

One incredibly exciting event was observed by the TBS flight on 26 July, when the balloon was parked around 800 m above the ground. As shown in Figure 10a, the aerosol total number concentration peak indicates the TBS aerosol payload sampled an aerosol plume from 17:59 to 18:12 UTC, which was apparently from a local biomass burning event. Note that due to the extremely high aerosol number concentration, both CPC and POPS were undercounting in this case, so the data presented here to demonstrate the event and for quantitative analysis, please use with caution. Figure 10b is zoomed into the period when the POPS caught the plume. The darker red color indicates a higher concentration of aerosols. We used computer-controlled scanning electron microscopy with energy-dispersive X-ray (CCSEM/EDX) spectroscopy to investigate the size, morphology and elemental composition of individual aerosol particles collected during this biomass burning event (examples are shown in Figure 10c). The above observations indicated both instruments observed the plume probably from the near site biomass burning event. Carbonaceous compounds dominate submicron particles at altitudes of 850 m and 610 m (one hour later). The example of a case with the ambient aerosol particles total concentration below the counting limit is shown in Appendix A, and the photo of the TBS deployed at the SGP is shown in Appendix A.

## 5. Conclusions

We have carried out the characterization of a commercial version of the POPS. The detection range of the commercial version of the POPS is the same as in the NOAA version (~135 nm to 3000 nm). The statistic shows that POPS has a stable counting efficiency among all versions and similar concentration counting limits as UHSAS (~3000 cm^−3^). This study mainly focused on the counting efficiency around the lower size range because the total number concentration measured by the POPS is more sensitive to the counting efficiency change of the 135–300 nm range. However, the counting efficiency of this POPS held close to 100% when we tested with 707 ± 9 nnm, 1019 ± 15 nm, and 2020 ± 15 nm PSL particles. The main improvements of the tested version include the temperature sensor in the aerosol flow pathway and the laser heater to keep the laser temperature above 35 °C. The lab study and the field deployment demonstrate that the data quality remains intact, while the laser temperature was above 25 °C. When sampling under humid conditions, the science driver for the study will dictate if an additional aerosol sample conditioner is required. For example, it is recommended to reduce the RH value to lower than 40% for dry size distribution measurement. Meanwhile, the Arctic deployment did not require an extra dryer as long as the enclosure temperature remained 15 °C above the ambient temperature.

Concentration comparison was carried out using the simultaneous sampling by POPS, SMPS and UHSAS. The results confirmed that POPS has the same detection limit as UHSAS. The coincidence error exceeds 25% when the total number concentration is larger than 4000 cm^−3^. Meanwhile, even with the total number concentration reaching 15,000 cm^−3^, POPS and UHSAS did not show significant size-shifting with the geometric mean diameters as demonstrated in Figure 3b. As expected, both chemical composition and aerosol particle morphology affect the POPS and UHSAS’s sizing accuracy because the size determination is based on either PSL or ammonium sulfate aerosol particles. The refractive index and particle morphology effect on the POPS are more profound for the POPS than that for the UHSAS.

A successful comparison flight with POPS and PCASP aboard was also achieved. Combining with the laboratory’s performance evaluation, we conclude that POPS can provide data of scientific quality. However, the size resolution of POPS is lower than UHSAS and PCASP at the broader size range and has a lower counting efficiency than PCASP at the lower size range.

Within the ARM program, the POPS has been deployed within the TBS suite of instruments at SGP and OLI sites between 2017 and 2020. The case study in 2019 demonstrated the value of the combined TBS dataset and suggested that the POPS equipped with a proper dilutor could contribute to the biomass burning study. TBS POPS has proven to be an essential measurement in the vertical assessment of atmospheric energy balance and, specifically, aerosol-cloud interactions.

## Figures and Tables

**Figure 1 sensors-20-06294-f001:**
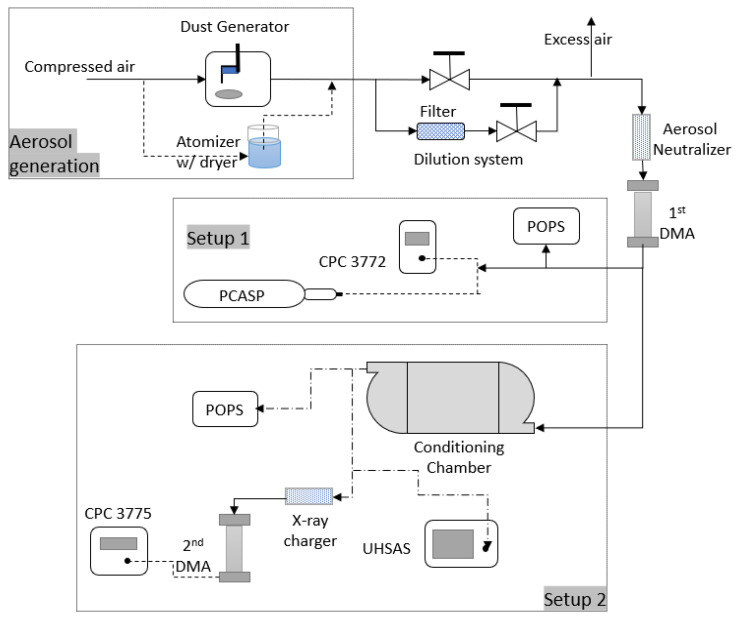
Diagram of the Portable Optical Particle Spectrometer (POPS) calibration setup.

**Figure 2 sensors-20-06294-f002:**
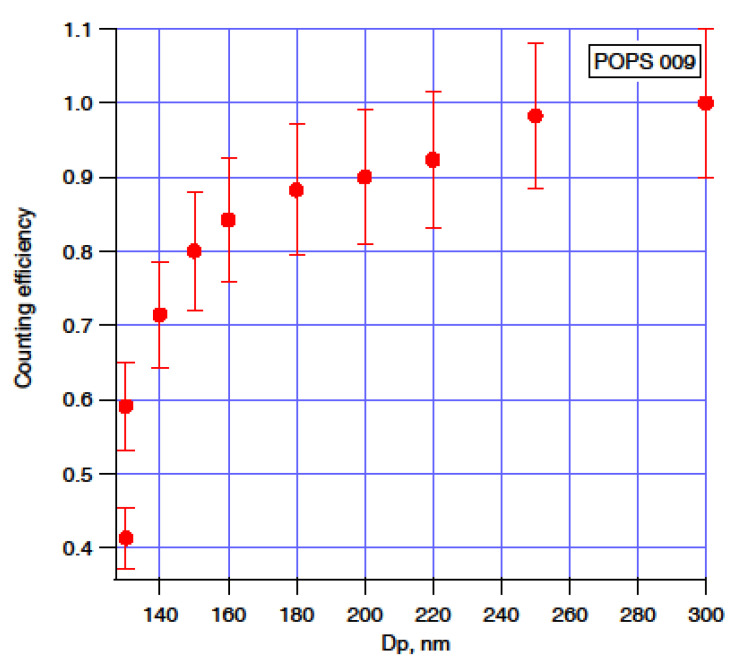
The counting efficiency of one POPS as a function of the particle (ammonium sulfate) electrical mobility size.

**Figure 3 sensors-20-06294-f003:**
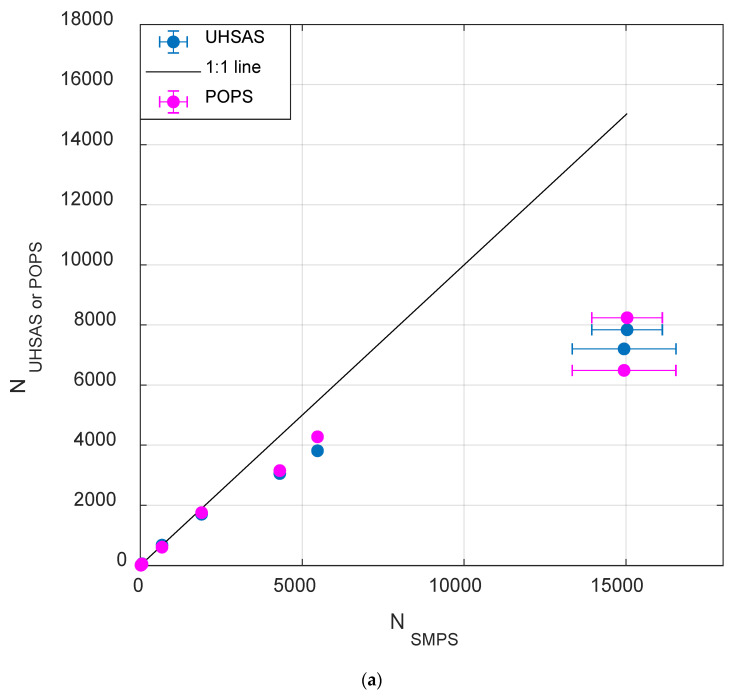
(**a**) The measured aerosol concentration comparison between Scanning Mobility Particle Spectrometer (SMPS), Ultra-High Sensitivity Aerosol Spectrometer (UHSAS), and POPS; the aerosol total number concentration effects on the (**b**) the normalized size distribution from POPS and UHSAS (using the 200 nm size-selected ammonium sulfate aerosols).

**Figure 4 sensors-20-06294-f004:**
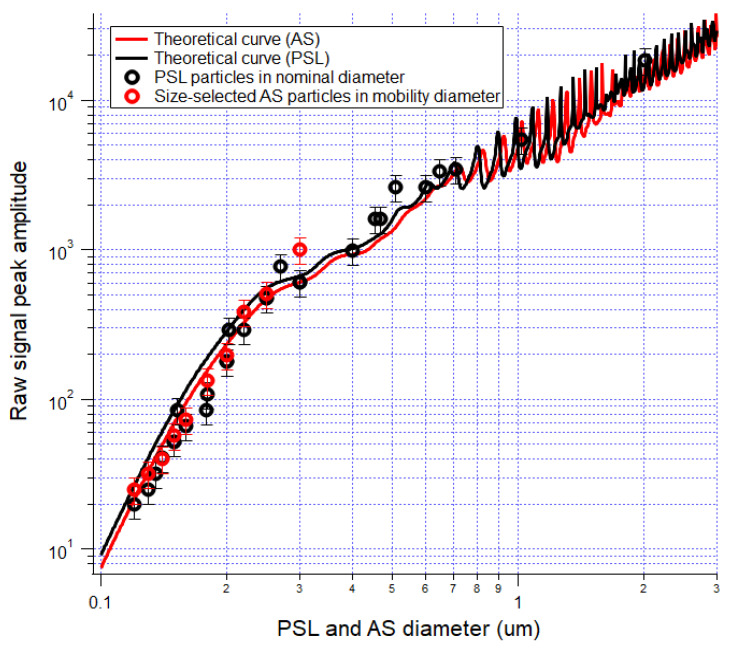
Calibration data of polystyrene latex (PSL) and ammonium sulfate particles. The scattering amplitudes for PSL and ammonium sulfate particles calculated using Mie theory as a function of particle diameter and scaled to the calibration results.

**Figure 5 sensors-20-06294-f005:**
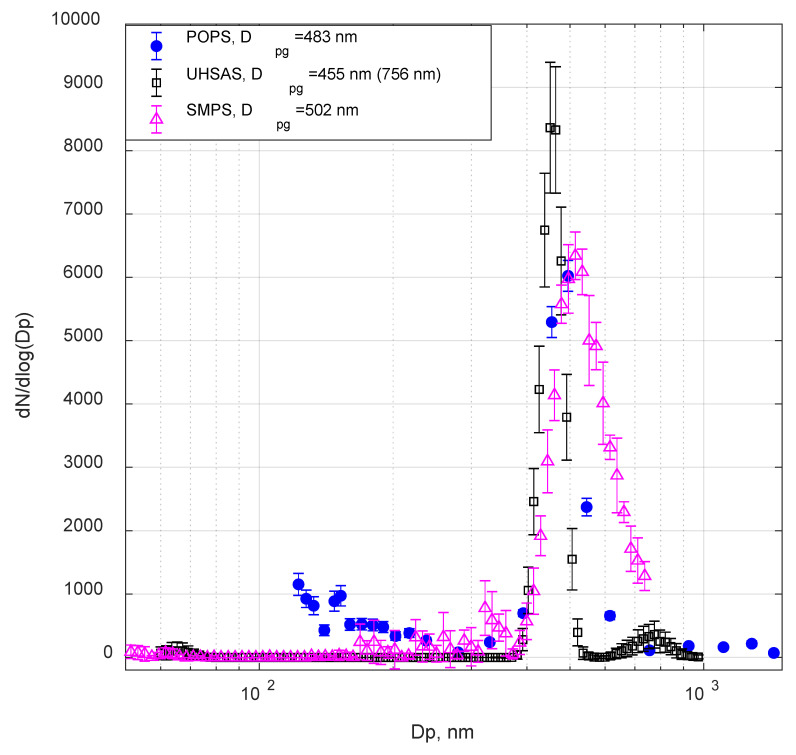
Size distribution comparison of 500 nm (electrical mobility size) ammonium sulfate particles.

**Figure 6 sensors-20-06294-f006:**
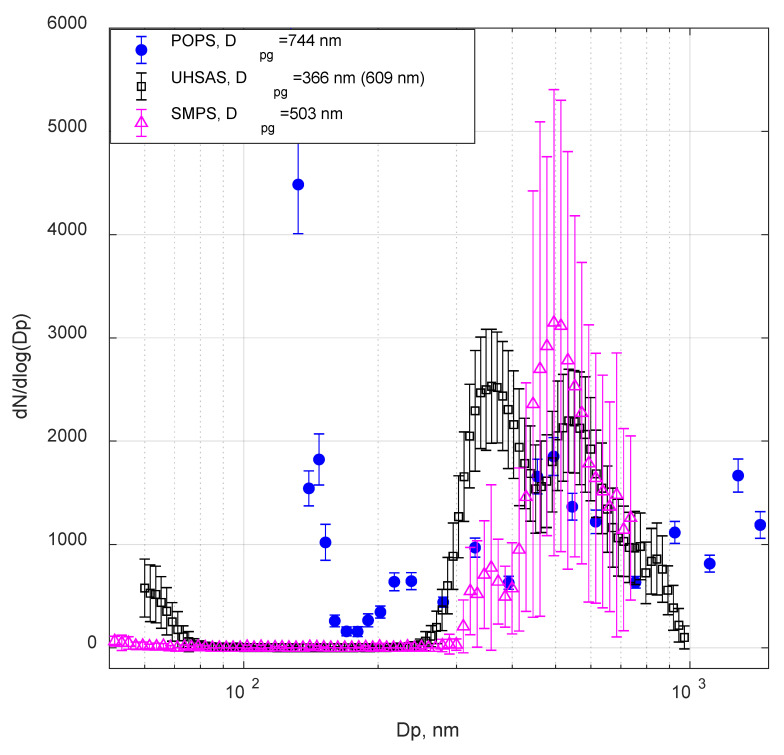
POPS and UHSAS size distribution of 500 nm (electrical mobility size) Arizona test dust (ATD) particles.

**Figure 7 sensors-20-06294-f007:**
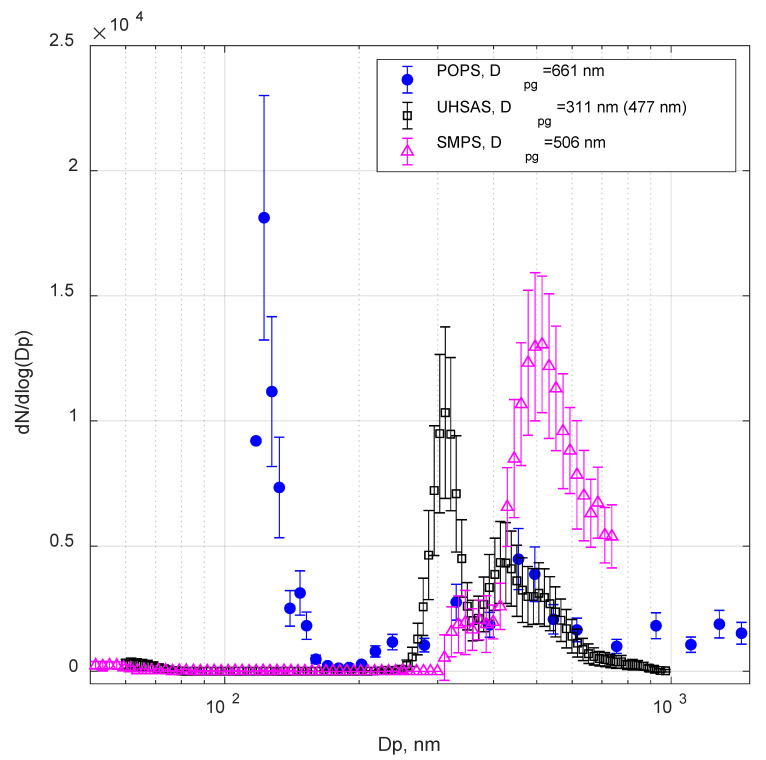
POPS and Passive Cavity Aerosol Spectrometer (PCASP) size distribution of 500 nm (electrical mobility size) titanium dioxide particles.

**Figure 8 sensors-20-06294-f008:**
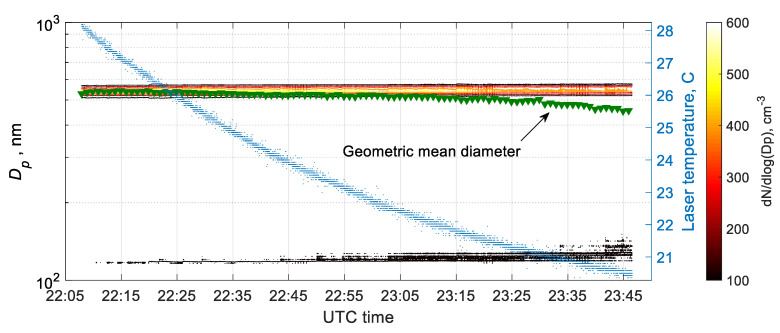
The optical laser temperature effect on the size distribution measurement for the 500 nm PSL aerosol particles.

**Figure 9 sensors-20-06294-f009:**
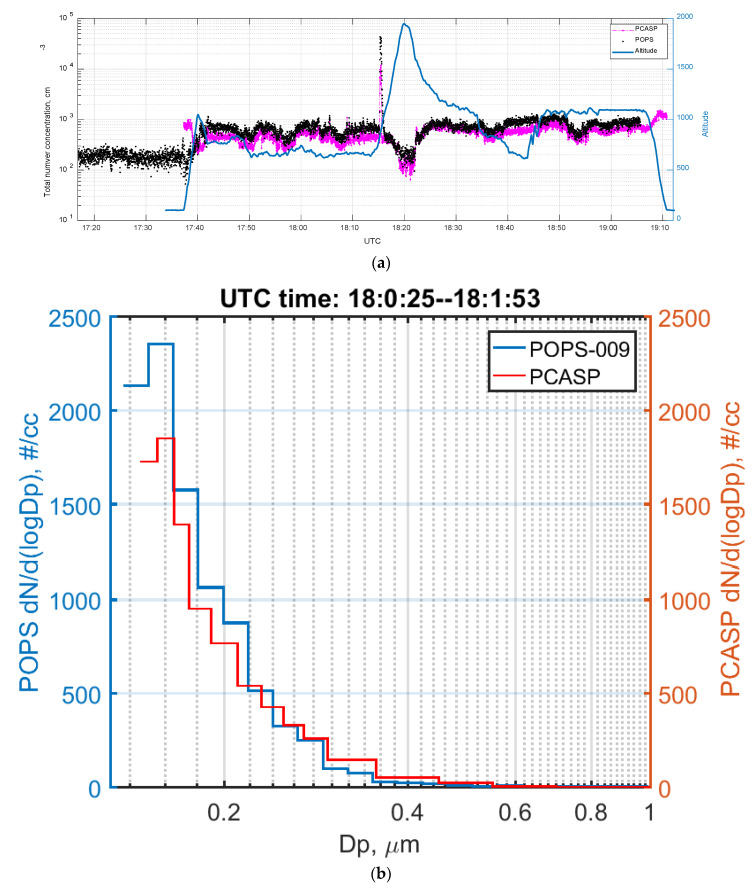
Comparison between POPS and PCASP during a flight on 17 August 2016: (**a**) Time series plot of POPS and PCASP; (**b**) size distribution of POPS and PCASP.

**Figure 10 sensors-20-06294-f010:**
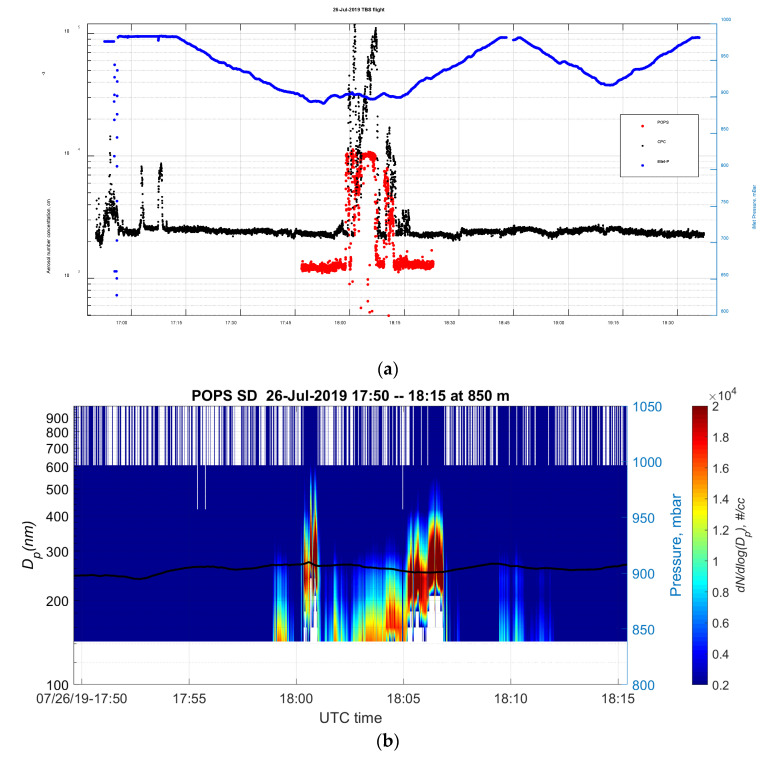
(**a**) The time series of the total concentration from the tethered balloon system (TBS) POPS and condensation particle counter (CPC) during the TBS flight on 26 July. (**b**) Aerosol size distribution plot of the biomass burning plume observed by the TBS POPS between 17:50 and 18:15 at 850 m. (**c**) Size-resolved chemical composition of particles collected between 17:50 and 18:15 at 850 m.

**Table 1 sensors-20-06294-t001:** Properties of tested aerosol particles.

Test Aerosol	Refractive Index at 405 nm	Morphology	Source	Reference
PSL	1.63 + 0.001i	Spherical	Applied Physics. Inc.	[23]
Ammonium sulfate	1.53	Slightly non-spherical	Sigma-Aldrich	[24,25]
Arizona Test Dust	1.51 + 0.00102i	Irregular	Powder Technology Inc.	[26,27]
Silicon dioxide	1.4696	Spherical	Sigma-Aldrich	[27,28]
Titanium dioxide	2.682	Aggregates of smaller size spheres	Aldrich	[29]

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
