# Peer review of "Performance Assessment of Portable Optical Particle Spectrometer (POPS)"

_sensors, 2020, doi:10.3390/s20216294_

Round 1
Reviewer 1 Report
The paper presents the evaluation of a new instrument in the laboratory and field tests. The work appears to me well performed and correct. Especially I regard the evaluation of the test dusts, including morphology effects, very valuable.
I think this manuscript should be published after some revisions, which are:
- 383. Maybe I missed something, but I think with the described set up the hysteresis of deliquescence and efflorescence cannot be shown, since it can only humidify before dried particles, not the other way round. Nevertheless, growth of course can be shown.
- The most likely detected small particles problem appears already in Fig. 3b but is only discussed later in the text.
- Can the judgement of the POPS sensor be made more quantitative? L. 515/516 in the conclusions are a bit weak.
- Therefore, figs. 3b and c might be merged or in another way the comparison should be made more clear. Maybe some statistical values of the distributions could be compared to make the deviations more quantitative. Also counting efficiencies (fig3a, fig 11a) could be discussed and/or compared with other typical instruments.
- The authors should accordingly adjust the conclusions section and the abstract.
- Fig 11a, could the authors comment on the overestimated small particle number with respect to possible false counts presented in fig. 8?
Some minor issues that should be corrected.
- Table S1 does not have units.
- The effect of multiple charges are only mentioned towards the end of the text (l. 298 and 313), although it appears already in Fig. 3a.
- 418/419 I think there is a problem with the sentence.
- In the supplement: What means "…with some monomers in clusters in the order of 743"?
Author Response
We sincerely appreciate the comments and suggestions from our reviewer. Thank you very much for considering the publication of our manuscript. We address your comments in the attached file (also in blue).

Reviewer 2 Report
Review on the Manuscript entitled: "Characterization and Operation of a Portable Optical Particle Spectrometer (POPS)"
The manuscript presents laboratory and field studies of an updated version of the Portable Optical Spectrometer (POPS), introduced by Gao et al. (2016), which relies on measurements of scattered light from aerosol particles for providing their size distribution in the range of ca. 140 - 3000 nm. While the POPS can be employed for ground observations, it has the potential to be employed on lightweight aerial platforms (e.g., Unmanned Aerial Systems; UASs and Tethered Balloon Systems; TBSs) for conducting atmospheric relevant observations on the vertical plane, due to its compact dimensions and light weight. Therefore the assessment of the performance of such an instrument is of great importance for ensuring that the instrument will provide accurate results during performing the above tasks (i.e., ground and aerial observations of aerosols size distributions). Besides the reliability of the hardware/software (important for any scientific instrument), the performance of the POPS mainly regards to its counting and sizing efficiency, similarly to other instruments which are based on the same operating principle. In this work the counting and sizing ability of the POPS were assessed by comparing its measurements to those of other instruments (i.e., condensation particle counter; CPC and Passive Cavity Aerosol Spectrometer Probe; PCASP, Scanning Mobility Particle Sizer; SMPS and Ultra-High Sensitivity Aerosol Spectrometer; UHSAS). Monodisperse aerosol particles of ammonium sulfate, Arizona Test Dust (ATD) and TiO2 were used for these purposes, while the response of the instrument (i.e., signal amplitude from scattered light) was obtained at different sizes by using Polystyrene Latex Spheres (PSL). In addition, the authors tested the performance of the POPS I) when its laser was operated under variable temperatures (i.e., ca. 30 to 20 ËšC) with monodisperse PSL particles and II) when the instrument was sampling air at variable relative humidity (RH; 40-90%), with monodisperse PSL and ammonium sulfate particles. Field applications including test flights with a manned aircraft and a TBS, when the POPS was sampling simultaneously with other instruments (i.e., PCASP and portable CPC, respectively) are also presented in this work and the results are discussed in a comparative manner. The authors conclude that the POPS has a comparable counting efficiency with an instrument operating under the same principle (i.e., UHSAS). Based on the above result and those obtained from the successful test flights and the comparison between the measurements obtained from the POPS and the PCASP the authors conclude that the instrument can be successfully fulfill its role for conducting aerial observations of aerosols size distributions. In addition, the authors underline the improvement of adding an active temperature control in the POPS laser (i.e., tested version of the instrument).
General Comments
This study is of great importance since the field of employing lightweight aerial platforms for conducting research flights of atmospheric relevance is continuously expanding following the evolution of miniature sensors/instruments. While this study is extensive by including comparisons of the POPS with other optical instruments (i.e., PCASP and UHSAS), as well as instruments that are considered reference (i.e., CPC and SMPS), when sampling aerosols of different chemical composition it has strong limitations and can be significantly improved. More specifically:
- a) The authors report the counting efficiency of the POPS using one type of aerosol (i.e., ammonium sulfate) in a narrow range of submicron particles (i.e., 135-300 nm), and not for bigger particles. This limits the credibility of the results (i.e., counting efficiency) to only a narrow range of sizes, in the lower end of the instrument's measuring range, as the POPS is intended to measure concentration of particles up to 2.5 μm The authors should complement the specific aspect of this work (i.e., counting efficiency) by showing and discussing results obtained when the instrument samples particles bigger than 300 nm or justify, by providing adequate evidence, that the counting efficiency they measured remains unchanged when the instrument samples bigger than 300 nm particles.
- b) The authors report the sizing efficiency of the instrument based on measurements with monodisperse aerosols of different chemical compositions (positive), however of only one size (i.e., 500 nm). As the POPS is intended to measure the size distributions of particles in the range of ca. 140 - 3000 μm, depicting and discussing the sizing efficiency of the instrument obtained only with one size is a strong limitation. The authors should complement the specific aspect of this work (i.e., sizing efficiency) by showing and depicting results obtained when the instrument samples particles of different sizes than 500 nm. Alternatively they should justify, by providing adequate evidence, that the sizing ability they measured remains unchanged when the instrument samples other than 500 nm particles.
- c) The main focus of this work is to evaluate the performance of an instrument with the potential to be employed onboard small UASs and TBSs. Both these aerial platforms lack, in most of the cases, pressurized compartments, resulting in the exposure of the instruments at pressures and temperatures lower than those prevailing on the ground, which may reach extreme values at high altitudes. While the authors claim that they evaluate the performance of the POPS at different ambient conditions (cf. introduction: "This paper focused on evaluating the variability of operational parameters between different versions of POPS, characterizing the operation envelope of ambient pressure..."), it seems that they did not tested the instrument for any possible effects related to differences in the absolute pressure. They only report in section 3.1, that the counting efficiency holds to a pressure difference between ambient and inlet of 150 hPa, without any further information on why is this happening. In addition, no information regarding the sizing performance of the POPS when sampling at reduced inlet pressure is reported. Note that, such a pressure difference between the instrument's inlet and ambient (pump outlet) will not be present during aerial observations with aerial platforms with unpressurized compartments (i.e., inlet and outlet will be exposed at nearly the same absolute pressure). However, on the other hand, it is possible that other phenomena, related only to the absolute pressure (i.e., and not the differential pressure between the instrument's inlet and outlet) could affect the instrument's performance. How are the authors able to support that in their work they characterized the operational envelope of ambient pressure, when they only tested the instrument at room/ground absolute pressure, inducing only a pressure difference of 150 hPa between its inlet and outlet? This is different than exposing the whole instrument to lower pressures, as will happen in reality during aerial observations with UAVs and TBSs. Did the authors performed any tests at lower absolute pressures than that of the room? Similarly, the authors report that the POPS were employed in the Arctic, exposed to temperatures significantly lower than 0ËšC, but they do not provide any performance related information. They only use this campaign for pointing out that a heated inlet and optical chamber result in lowering the RH of the aerosol, thus effectively drying it. It is not clear however if the instrument's performance remained unchanged under these extreme temperature conditions.
- d) Certain aspects regarding the presentation and discussion of the results can be majorly improved in order to provide information to the reader in a clearer and more quantitative manner. For instance, in addition of depicting the measured size distributions (figures 3b and c, 5, 6 and 7), the authors can calculate and provide the geometric mean diameter, which is more quantitative than the peaks in the figures, and then compare it with the nominal size of the particles. This will benefit the discussion in the relevant sections and in the conclusions by providing quantitative results. As an example, what is the sizing efficiency of the POPS when sampling 500 nm of ammonium sulfate particles? By how much is the sizing efficiency affected when the POPS samples dust particles of the same size? It is difficult for the reader to answer these very important questions by looking at the figures or by reading the text.
- e) Similar to the above comment, the presentation and discussion of the results when the POPS laser was operated at different than its nominal, temperatures (i.e., section 3.5) can be improved. For example in figure 8 the reader can observe an increase in the particle counts at the smaller size bins (i.e., ca. 140 nm) when the temperature of the laser is reduced below 25 ËšC, but it is not very clear if the signal of PSL spheres at approx. 500 nm remains undisturbed. In addition, there is no available information from the figure and the discussion on the effects of temperature in the counting efficiency of the instrument (e.g., how the total number concentrations were affected).
- f) In section 3.6 the authors discuss the effects of the sample RH on the performance of the POPS, which is one of the positive aspects of their work. They suggest that maintaining the temperature of the optical chamber higher than that of the environment results in reducing the RH in which the aerosols are exposed to, during their measurement and provide calculations for proving it. Then they use these calculations as a guideline for evaluating and interpreting field data, which is reasonable. They provide two examples of field campaigns where the outside/ambient air temperatures varied from -25 to +15 ËšC and from 0 to +40 ËšC, while the instrument's optical chamber was maintained at a temperature difference (i.e., between the optical chamber and the environment) of 15 and 20 ËšC, respectively. How was the above temperature differences verified?
Based on the calculations/suggested guideline they conclude that the RH in which the particles are exposed to during their measurement, resides <40% (i.e., sampling under dry conditions), therefore no measuring artifacts induced by increase RH conditions are expected. However, a temperature difference between the optical chamber of the POPS and the environment of 15 ËšC (i.e., in the first example) results in an temperature inside the optics ranging from -10 to 30 ËšC. In section 3.5 the authors report a degradation of the instrument's performance when the laser temperature is lower than 25 ËšC. Was the temperature of the laser always higher than 25 ËšC even when the temperatures of the optical chamber were significantly lower (e.g., at -10 ËšC)? Does the POPS have the ability to measure the temperatures of its optical chamber and the laser independently?
- g) The results and the main, "take home" messages can be significantly improved by removing or compacting some secondary information and by including quantitative information regarding the main aspects of this work. For example the phrase "The detection range of the commercial version of the POPS is the same as the NOAA version (~135 nm to 3000 nm)." is not of the biggest importance. This information could still be present in the conclusions (if the authors consider it important) but in a more compact way, by combining it for example with the first sentence of the conclusions. On the other hand, the reader is informed that the POPS has a stable counting efficiency but without any quantitative information on how much this counting efficiency is, in respect to the reference and/or other instruments used in this work. No information on the sizing performance of the instrument in a clear manner are reported in the conclusions. On the contrary the "take home" message is that the POPS has a lower size resolution than the UHSAS, which is not that important considering that these instruments are measuring accumulation and coarse mode particles (i.e., maximum two modes, at least in the atmosphere), which can be resolved by fitting lognormal distributions even at 6 size bins. In addition, the authors claim in the conclusions that "the POPS contributed to the biomass burning study", but this contribution is not clear neither in the conclusions nor in the discussion. The ability of an instrument to probe particles and register values during a biomass burning event alone, does not contribute in expanding our knowledge. This is especially true for an instrument which counting accuracy is significantly reduced at high particle number concentrations, showing an order of magnitude (i.e., approx 90000) less particles than the (reference) CPC. In addition to that, there are no solid evidence that the reported by the instrument size distribution and apparent growth is accurate.
- h) The abstract could briefly incorporate some of the main messages of this work in a quantitative manner. For example, it could include the accuracy of the instrument in terms of the counting/sizing efficiency, compared with reference/other instruments used in this study and/or how this accuracy is affected by the different conditions (i.e., RH, pressure, temperature). Also the sentence: "The main goal of the current work is to provide practical guidance on using the POPS for the in-situ airborne measurements." seems irrelevant to the nature of this work. In more detail, this work regards the assessment of the instrument's performance under different conditions, like sampling of aerosols having chemical composition that differs from the one that the instrument is calibrated with, wide range of particle number concentrations, varying temperature and RH. While some "operational tips" are provided in the text, I do not see any particular reason for reporting that the main goal of this manuscript is to provide operational guidance.
Major specific comments
1) The title can be improved in order to i) clearly distinguish this work from that of Gao et al., 2016 and ii) for emphasizing on the main positive aspects of this work (e.g., different types of aerosols used). On the contrary, the word "operation" used in the title does not seem relevant to this work. In addition, the word "characterization" in the title is misleading and not supported by the results presented in the current version of the manuscript (i.e., narrow range of sub-300 nm monodisperse particles for assessing the counting efficiency and only one monodisperse size for assessing the sizing efficiency of an instrument capable of measuring particles up to 3000 nm). The authors should complement their work providing additional evidence and/or adequate justification that the measured counting and sizing efficiency remains unaffected when the instrument samples particles of different sizes (i.e., representative of its full sizing range) than the ones used in this work. Alternatively they should modify the title in order to better reflect exactly the work described in this manuscript (i.e., tests with limited particle sizes but of different chemical composition).
2) Abstract (line: 20): " This study evaluates the performance of the POPS under different environmental conditions...".
It would be good to clarify and precisely mention what the authors mean with the term "different environmental conditions".
3) Abstract (lines: 21-24): "Results of comparison between POPS and several other sensors, including its full-size airborne ‘sibling’ instruments – Passive Cavity Aerosol Spectrometer Probe (PCASP) and Ultra-High Sensitivity Aerosol Spectrometer (UHSAS), are reported."
It would be preferable to include some quantitative results in the abstract regarding the above mentioned comparison (cf. general comment #g).
4) Abstract (lines: 24-25): "The main goal of the current work is to provide practical guidance on using the POPS for the in-situ airborne measurements."
How is this "main goal" justified? In a manuscript with 11 figures depicting the performance of POPS in comparison with other instruments and with a text expanding over 20 pages the "practical operational guidelines" are limited only to a few sentences, like for example the temperature setpoint of the laser, the use or not of a dryer, depending on the temperature/RH conditions. There are not "practical guidelines" regarding the nature of sampled particles, the actual number concentration etc. (e.g., if the instrument samples dust at highly polluted environments then expect a deterioration of counting performance by X%). If the authors main intention is to provide operational guidelines for the POPS then they should emphasize on these matters as well, by utilizing their results and most importantly to clearly underline their findings in the conclusions.
5) Introduction (lines: 67-69): "This paper focused on evaluating the variability of operational parameters between different versions of POPS, characterizing the operation envelope of ambient pressure, environmental temperature, and sample relative humidity."
- i) How the variability of the operational parameters between different versions of the POPS is evaluated? From the manuscript it is clear that the only operational parameter between the tested POPS systems that was modified/changed was the laser active temperature control.
- ii) Where exactly in the manuscript (besides one sentence in section 3.1) is the operational envelope of ambient pressure discussed? Where are the supporting measurements/evidence for corroborating that the "operation envelope" of ambient pressure was studied/characterized?
iii) The authors use the term "envelope of environmental temperature". However, they only tested the performance of the instrument, with its laser heater off (i.e., active temperature control off) at zero degrees (i.e., ambient temperature) for evaluating the effects of temperature on the instrument's performance. While their findings and consequent recommendation for enabling the active temperature control are important for reassuring an acceptable performance of the instrument, did they tested how effective their system is at temperatures relevant to those of high altitude\polar exposure (i.e., < -15ËšC)? I believe that this question is well justified since this instrument is intended to be used with TBSs (very low temperatures at high altitudes) and in polar regions. In fact, the authors mention (section 3.6, lines 405-406) that the ambient temperature in "one Arctic observation site data between April to October, the ambient temperature and RH at the ARM mobile facility (AMF, Oliktok Point, AK) is between -25 to 15 ËšC, and 20% to 95%, respectively." and that during that campaign the POPS optical chamber was maintained at 15 ËšC above the ambient temperature. This results to an operating optical chamber temperature envelope ranging from -10 to 30 ËšC. The authors, have tested their optical system down to ca. 20 ËšC (having an ambient temperature of zero degrees), reporting counting performance deterioration (at least to the lower size bins) below 25 ËšC. Therefore, shouldn't the performance of the instrument be tested also when the ambient temperature is at -25 ËšC, (or when the optics are exposed to -10 ËšC, as probably happened), since this instrument was exposed at these temperature conditions in order to be able to justify a study throughout the operating environmental temperature envelope?
5) Section 2 (lines: 173-176): The authors describe the use of a conditioning chamber for varying the RH of the sampled aerosols. No description on how the chamber was used for varying the temperature of the sample stream is provided (if that was the same chamber used for experiments described in section 3.5).
6) Section 3.1 (lines: 189-190): "We have found that the counting efficiency curve shown in Fig. 2 holds when the difference between the ambient pressure and the inlet pressure is less than 150 hPa."
This is the only time that the effects of pressure on the instruments performance (here limited to counting efficiency, only) are reported in the manuscript. On the contrary the authors declare (abstract and introduction) that i) the POPS is designed for usage onboard lightweight aerial platforms (i.e., UASs and TBSs), resulting in its exposure to ambient pressures lower than those prevailing at the ground/sea level and ii) that they characterized the POPS operational envelope in terms of ambient pressure. Can the authors provide additional measurements/evidence than those mentioned in the above sentence in order to support these aspects of their work? In addition, increased differential pressure between the instrument's inlet and outlet can have different effects (e.g., reduced flow due to insufficient pump power at high pressure difference) than those of exposing the whole instrument at reduced absolute pressure conditions (i.e., similar to pressure conditions encountered during aerial research flights with UAVs and/or TBSs). Is this aspect (i.e., exposing the POPS at absolute pressures lower than those prevailing at the ground) accounted for in this work?
7) Section 3.1 (line 180) : The title of the section is "Counting efficiency", however it provides results of POPS counting efficiency for particles ≤300 nm. It is not evident, neither from figure 2, nor from the discussion if the instrument's counting efficiency remains close to 100% when it samples bigger particles, especially since the POPS measuring range reaches sizes of 3000 nm. The authors should complement this section by depicting and/or discussing the counting efficiency of the instrument when it samples bigger particles as well. Else, what they discuss in this section and depict in figure 2 is more resembling an attempt to define the D50 (i.e., size where 50% detection efficiency is reached) with ammonium sulfate particles, which is not different than the procedure that the manufacturer follows (cf. figure S1).
8) Section 3.1 (lines 205-208): "The average standard deviation of the baseline measured for 205 these instruments was 9.3 ± 2.1 digitized counts (noise in the raw signal), while the average baseline 206 value was 2170 ± 110 digitized counts. These values translate to a minimum detectable raw particle peak 207 signal of 27 A/D counts, which corresponds to a PSL diameter of approximately 110 nm."
- i) It is not clear from these lines to which measurements the authors refer to. Do they refer to the measurements conducted in this work or in the measurements conducted by the manufacturer as mentioned in the previous sentence?
- ii) It is not clear from these lines (but also from the rest of the text in section 3.1) what was the defined by this work D50 and how it is compared to the value provided by the manufacturer.
9) Section 3.1 (lines 211-212): "The inlet and exhaust ports of the instrument under test have been modified as described above."
It is not clear to which modifications the authors refer to. Where is the modification of the inlet/outlet described? If described in details in another section then please direct the reader to that section. If not described in the manuscript, then please provide this description.
10) Section 3.2 (lines 229-232): "Comparison of the aerosol size distributions at two total concentrations (around 700 cm-3 and 15,000 cm-3) measured by POPS and UHSAS did not show significant size-shifting (Figure 3b and 3c). The shape of the size distributions remained practically the same, although both UHSAS and POPS experience a significant undercounting issue at 15,000 cm-3."
- i) The shape of the size distributions does not dictate that geometric mean diameter of measured size distributions will remain the same (i.e., affecting the sizing ability of the instrument). This is important since in most aerosol studies the geometric mean diameter is used for expressing the size of particles or the evolution of any size-related phenomena. From figures 3b and c it seems that the geometric mean diameter shifts towards bigger sizes, when the instruments sample at 15000 cm-3. The authors can calculate the geometric mean diameters from the distributions they measured at low (i.e., 700 #/cm3), high (i.e., 15000 #/cm3) and at intermediate (i.e., 3000 - 4000 #/cm3) particle number concentrations. In this manner they will obtain quantitative results, significantly contributing in the discussion and in their conclusions.
- ii) More than one peaks are visible in figures 3b and 3c, while the instruments sample monodipserse particles having an electrical mobility diameter of 200 nm. The peaks are visible even when the instruments are sampling particles at relatively low particle number concentrations, with the dominant peak to be around 200 nm, as expected, and other peaks present at 350 and 450 nm for the POPS and at 300, 400, 500 and 600 nm for the UHSAS. Can the authors explain why when sampling monodisperse particles both the UHSAS and the POPS show distinct peaks? Is this related to multiple charged particles introduced by the DMA? Interestingly, similar distinct peaks at almost the same sizes are visible during simulations for the POPS response when measuring ammonium sulfate particles with similar size (cf. Figure 8 in Gao et al., 2016). So is it possible that these distinct peaks at sizes different than that of the monodisperse ammonium sulfate particles can be attributed to the instrument itself (e.g., calibration method, Mie scattering undulations, other reasons)?
The authors should comment on this and an estimation on the number and real size of multiply charged particles (originating from the DMA) together with a comparison with their observations can be helpful.
iii) Please define "SD" (i.e., y-axis title in figures 3b and c).
11) Section 3.2 (lines 263-264): "Using the characterization setup 2 (Figure 1), we also compared the responses from POPS, UHSAS and SMPS for the size-selected ammonium sulfate aerosol particles."
The size of the monodisperse ammonium sulfate particles with which the tests were conducted is not mentioned, making it difficult for the reader to understand, without looking at the corresponding figures.
12) Section 3.3 (lines 274-275): " Both measured sizes (equivalent optical diameter) by POPS and UHSAS are smaller than the SMPS size (mobility diameter).
How the "measured sizes" were defined in this case? From the dominant peak of the size distributions, as geometric mean diameters by any other means?
13) Section 3.3 (lines 277-278): "Note that the increase in the number of the POPS size bins may cause the sizing issue due to the POPS complex size determination, as shown in Figure S7(b). "
Which sizing issue the authors refer to? This sentence needs clarification and better explanation.
14) Section 3.3 (Figure 5): In addition of depicting the measured size distributions from the instruments, the geometric mean diameters can be calculated, resulting in enriching the discussion with quantitative results (cf. general comment #d).
15) Section 3.4 (lines 296-298): "A secondary peak at the larger sizes on the distributions is the usual artifact of electrical mobility based separation systems; it is formed by the multiple charged particles passed through the DMA column."
It is not clear to which instrument(s) the authors refer to. In addition, this sentence can be supported and enriched by calculating the amount of multiple charged particles and their actual/real size when classified by the first DMA of the setup.
16) Section 3.4 (lines 307-319): The authors make an attempt to explain all these "additional" peaks in figures 6 and 7. In more detail, in figure 6 the POPS exhibits two peaks at sizes > 700 nm and two peaks at sizes < 500 nm (i.e., at ca. 250 and 350 nm), excluding the first 4 size channels, while the dominant peak is at approx. 500nm. The UHSAS on the other hand exhibits two peaks of comparable amplitude, excluding the lowest size channels, one at ca. 350 and another at 500 nm. In figure 7 the POPS exhibits one peak at sizes > 700 nm and two at ca. 250 and 350 nm, excluding all the size channels <200 nm. The dominant peak seems to be around 450 nm. On the other hand the UHSAS exhibits a dominant peak at ca. 300 nm and a second peak of lower amplitude but wider at ca. 400 nm.
- i) They attribute the response of the optical instruments (i.e., resulting in additional peaks) to the shape of the aerosols and different optical properties. While both are true and probably have contributed significantly in the results obtained from both the POPS and the UHSAS, the presence of additional peaks at almost the same sizes when the POPS sampled monodisperse 500 nm ammonium sulfate particles (i.e., a single refractive index, shape factor close to spherical particles) is concerning. Is it possible that the multiply charged particles may have played a role in the different observable peaks, or this can be also related to the optical instrument(s) response when sampling particles with different optical properties than those calibrated with, similar to Gao et al., 2016 (cf. also comment 10 ii)?
- ii) The discussion in the whole section 3.4 can be improved if additionally to depicting the measured size distributions, the geometric mean diameters were also calculated for the different types (i.e., chemical compositions of aerosols). In this manner, the discussion will be enriched and it would be possible for a potential user of the POPS to quantify the associated uncertainty when measuring aerosol particles of the mentioned chemical compositions. For example, in most atmospheric observations the size distributions measured by optical instruments are treated as having maximum 2 modes (i.e., accumulation and possibly coarse). Therefore it is common to express these size distribution measurements in terms of geometric mean diameters and particle number concentrations of each mode. So, despite if an instrument exhibits adjacent peaks in the sub-micrometer size range, one geometric mean diameter can be assigned in this size range. This, helps in smoothing the measurements as well. So it would be beneficial for this study to include similar results and discussion in this section.
17) Section 3.6 (lines 386-388): " The maximum growth factor for POPS and UHSAS (around 1.21) is smaller then SMPS derived one (1.54), which is consistent with the other growth factor study with the differential mobility technique [46]."
- i) The "growth factor" should be defined. It would be also preferable to be mentioned as "hygroscopic growth factor" .
- ii) It is not clear what the meaning of this phrase is. In addition the authors cite a paper written in German (i.e., Magnus, G. J. A. d. P., Versuche über die Spannkräfte des Wasserdampfs.; cannot be understood by non German speakers), which most impotently dates back in 1844! During that period the differential mobility technique for sizing/classifying aerosols had not been discovered yet.
If the authors are trying to say that the hygroscopic growth factor obtained from the optical measurements is smaller than that obtained from the electrical mobility measurements, then they have to write it in a more clear way, while providing appropriate reference(s).
iii) The hygroscopic growth factor of ammonium sulfate particles with a dry electrical mobility diameter of 200 nm (i.e., at <40% RH), should be in the order of 1.7 when these particles are exposed to 90% RH (Seinfeld and Pandis, 2006). This value has been confirmed with methods which are based on differential electrical mobility techniques, namely the Hygroscopic Tandem Differential Mobility Analyzer (HTDMA; Rader and McMurry, 1986). So the value of 1.54 that the authors observed is by no means consistent with the available literature and the scientific observations. This can be most probably explained by the fact that the sheath of the second DMA used in their experiments (i.e., denoted as SMPS in the above sentence) had RH levels that were significantly lower than the 90% RH of the aerosol sample, resulting in changes of particles size inside the classification column (Stanier et al., 2004; Biskos et al., 2006). Although that this work is not focused on particle hygroscopicity, the authors should be precise when reporting results and compare them with other scientific observations.
18) Section 3.6 (lines 394-396): "There several ways to control the relative humidity of the aerosol samples. The most straightforward approach is via change of the temperature of the aerosol inlet flow".
While heating the aerosol inlet flow will reduce the RH in which the aerosol is exposed to, thus "drying" it, this method does not guarantee that the RH will remain low during the measurement of the aerosol. In more details, if the temperature drops back to ambient levels prior of the aerosol measurement (i.e., within the instrument) then the RH in which the particles will be exposed to, will increase. In other words, if the temperature is not maintained to higher levels than that of the ambient inside the optical system of the instrument, then water will re-condense on the particles and possibly on other solid surfaces inside the optics. So it is not straightforward to simply heat the aerosol inlet. Heating the aerosol inlet and the optics, which is probably the case for the instrument used in this study, sounds a more plausible approach for reducing the RH and consequently the water content of the sampled aerosol. The authors should clarify if they only heated the aerosol at the instrument's inlet or if they heated both the inlet and the optical chamber, as mentioned in the same section (lines 407 and 411) and how they measured these temperatures.
19) Section 4.2 (lines 478-480): " Note that due to the extremely high aerosol number concentration, both CPC and POPS were undercounting in this case, so the data presented here to demonstrate the event and can’t be used for quantitative analysis."
While the POPS is significantly underestimating the particle number concentration of particles due to coincidence (cf. section 3.2 of the manuscript) the TSI 3007 portable CPC has an upper counting threshold of 105 #/cm3, according to manufacturer (TSI; specifications sheet). Even if one accepts that close to the above maximum concentration value the instrument provides the particle number concentration with reduced accuracy, this extreme value is only reached twice during the first minutes around 18:00 hours (cf. Figure 12a in the manuscript). With the exception of these minutes, during which the CPC was recording extreme high values, the rest of the dataset can be used for calculating the POPS counting efficiency when exposed to biomass burning aerosols at high particle number concentrations. In addition the above sentence is misleading regarding the performance of the CPC, with the exception of a few minutes during which indeed the number concentration reached the maximum (i.e., 105 #/cm3).
20) Section 4.2 (lines 481-482): " We found the rapid growth of the biomass burning aerosols from 100 nm to 300 nm in a few minutes."
Here the authors seem to trust the measurements of the POPS in terms of sizing, while a few sentences above they claim that the data are used for demonstrating the event (cf. comment #19). Is there any particular reason for trusting the sizing ability of the POPS when exposed to high particle number concentrations when a) the counting ability is significantly deteriorated (also mentioned by the authors) and b) when the sizing performance of the instrument is not assessed at high particle number concentrations and when the instrument samples biomass burning originated particles (i.e., different optical properties from the aerosols tested in this work)?
- i) The authors should comment on this and adequately support it. If this (i.e., sizing performance does not deteriorate when the POPS is exposed to high particle number concentrations) is true, then it is a very useful result/conclusion.
- ii) On the other hand, could the apparent particle growth that the authors report and depict in Figure 12b, be the result of a measuring artefact, caused by the high particle number concentrations or by the different chemical composition/optical properties of the sampled aerosol?
21) Conclusions (lines 505-506): "We have carried out the characterization of a commercial version of the POPS. The envelope of POPS operation conditions is determined."
These conclusions are vague and probably misleading.
- i) The characterization of the performance (sizing/counting ability) of an instrument with a capability of measuring the size distribution of particles ranging from ca. 0.1 - 3 μm should include results covering (or being representative of) the above size range. An exception of the above, regards instruments that have been previously characterized throughout their full measuring range, something however that should be explicitly mentioned with the appropriate reference(s). According to my knowledge and understanding, the POPS sizing/counting performance has not been previously characterized throughout its full size range. Gao et al., 2016 (also cited in this work) performed a comparison with instruments capable of providing particle size distributions (SMPS, APS), under real operating conditions, but without discussing in detail the performance of the instrument during these comparison. In this work, the counting efficiency of the instrument was assessed in the range 0.15 - 0.3 μm and its sizing performance with particles of 0.5 μ In addition, there is not a clear justification on why these (limited) sizes were selected and on why one should expect that both the sizing and counting performance of the instrument should remain unchanged throughout its measuring range. The authors should either complement their results with those obtained with other sizes, representative of the instrument's measuring size range, or make use of a different term (i.e., than "characterization") throughout their manuscript to describe their work. In the second case (i.e., maintaining the current results without any additions), they should justify their selection on using the specific size(s) of particles for their experiments and focus on the effects of sampling particles with different chemical compositions.
- ii) Similarly "The envelope of POPS operation conditions" is not fully determined, accounting that this instrument is intended to be used onboard lightweight aerial platforms. For example, what is the maximum altitude onboard an unpressurized\unheated compartment (e.g., UAVs and TBSs) that this instrument can be used without a significant deterioration of its performance? What is the minimum temperature that the instrument can be exposed to, without significant performance degradation? Note that, the authors report that the POPS were employed in a campaign in the polar region, where the temperature reached -25 ËšC, during which the optics chamber temperature was -10 ËšC. While they make use of this campaign as a case study for point out that by heating the aerosol inlet and the optical chamber they effectively reduce the RH in which the particles are exposed to, they do not provide any details regarding the performance of the instrument at these temperatures. In addition, in their work they tested the instrument at 0 ËšC but with the laser heater off in order to study the effects of lower temperatures of the laser. However in the manuscript it is not clearly mentioned and justified by appropriate evidence, the minimum operating temperature of the instrument (without significant loss of data quality).
22) Conclusions (lines 507-508): "The statistic shows that POPS has a stable counting efficiency among all versions and similar concentration counting limits as UHSAS (~3000 cm-3). "
The counting efficiency of the POPS reduces above 3000 #/cm3, as demonstrated with monodisperse ammonium sulfate particles (i.e., 200 nm) and most probably can be further corroborated from the measurements during the biomass burning event. That part of the above sentence is well justified. On the other hand, the stability of its counting efficiency is not strongly supported as it was determined only in the limited size range of 135 to 300 nm and/or only with 200 nm ammonium sulfate particles. In addition the effects of aerosols with different chemical compositions on the instrument's counting efficiency are not reported. The authors should make use of the experiments conducted with other aerosol types (i.e., ATD, TiO2, PSL) and preferably to complement their work with additional experiments with bigger than 500 nm particles in order to justify that the POPS counting efficiency is stable (at least up to 3000 #/cm3).
23) Conclusions (general; refer also to my general comments):
- i) There are not conclusions regarding the sizing performance/efficiency of the POPS.
- ii) There are not conclusions on the pressures/altitudes that the POPS can be operated, while maintaining an acceptable performance. This is very important for the scope of this paper and the intended use of the instrument above aerial platforms.
24) Conclusions (lines 521-522): "The case study in 2019 demonstrated the value of the dataset that POPS contributed to the biomass burning study."
It is not clear from the manuscript how the POPS contributed to the biomass burning study, especially since its counting efficiency was deteriorated and no strong evidence are provided for supporting that its sizing ability was not affected, as well (cf. comment #20).
Minor specific comments
25) Introduction (lines 33-36): "High-quality measurements with suitable temporal and spatial resolutions are essential for addressing the research questions in meteorology, atmospheric processing, severe weather monitoring, and in many other areas of human activity, like industrial hygiene, semiconductor and pharmaceutical industry [4-7]".
The references used in this sentence are mostly related to UAV developments in general and with specialized instruments developed for UAVs. Since the sentence does not refer to aerial applications/observations (i.e., some applications can be as well ground based, while others are exclusively ground based) better and more representative citations can be used.
26) Introduction (lines 36-39): "A high-resolution real-time technique is critical to capture aerosol properties and their space and/or time variations, especially from a fast-moving platform (manned or unmanned). Due to its rapid evolution in the atmosphere, it is critical to capture and monitor the temporal and spatial variations of aerosol properties with a high time-resolution techniques."
Two sentences with the more or less the same meaning. One can go, or they can be combined. In addition, " with a high time-resolution techniques" should be corrected to " with high time-resolution techniques".
27) Introduction (lines 39-41): "Optical particle spectrometry (OPS) – a technique based on elastic light scattering – has served as a foundation for such a real-time 40 measurement for a few decades. [8-10]"
The authors probably refer to lightweight/compact applications, where indeed the OPS technique is commonly used. However, for ground based applications, where size/weight is not an issue, there are a lot of online techniques for characterizing the different aerosol properties (i.e., not only the size/optical properties). The authors should clarify if they refer to aerial/lightweight platforms within this sentence or prior of this sentence.
28) Introduction (lines 42-43): "Various OPS based sensors have been developed, and their performance, errors, and uncertainties were evaluated.[10-12]"
Reference Number 10: "Miles, et al., Sources of Error and Uncertainty in the Use of Cavity Ring Down Spectroscopy to Measure Aerosol Optical Properties. Aerosol Sci Tech 2011, 45, (11), 1360-1375." refers to an instrument that measures light extinction (i.e., sum of light absorption and scattering). In the sentences above the authors describe Optical particle spectrometry to be based on light scattering. So the reference here is not the most appropriate. The authors should think of replacing with a more appropriate one like for instance: " Szymanski, W. W., Nagy, A., Czitrovszky, A., Optical particle spectrometry - Problems and respects. Journal of Quantitative Spectroscopy & Radiative Transfer 110 (2009) 918–929. or another adequate citation.
29) Introduction (lines 43-45): " Several commercial OPS instruments emerged to provide a portable personal monitoring solution in the semiconductor and pharmaceutical industries, such as Grimm optical particle counters (OPC, models 1.108 and 1.109).[13]"
Since the authors refer to the Grimm models in this sentence it would be better to use the reference (11) from above:
Burkart, J.; Steiner, G.; Reischl, G.; Moshammer, H.; Neuberger, M.; Hitzenberger, R., Characterizing the performance of two optical particle counters (Grimm OPC1.108 and OPC1.109) under urban aerosol conditions. J Aerosol Sci 2010, 41, (10), 953-962.
here and move the more generic reference (13) Yoo, S. H.; Chae, S. K.; Liu, B. Y. H., Influence of particle refractive index on the lower detection limit of light scattering aerosol counters. Aerosol Sci Tech 1996, 25, (1), 1-10.
in the above sentence.
30) Introduction (lines 62-63): "The compact optical particle spectrometer (COPS) model features a hardened weatherproof enclosure for TBS deployment."
Please define the abbreviation "TBS" in the introduction during the first time it is used and not only in the abstract or in line 66.
31) Section 1.2 (lines 87-89):" The NOAA statistical analysis of the uncertainty on hypothetical aerosol populations indicated that the errors are less than 1.5% for the accumulation mode parameters and number concentration for the aerosol sizes between 140 and 3000 nm."
Please clarify the meaning of "accumulation mode parameters". Do the authors mean the parameters of the lognormal distribution (i.e., geometric standard deviation, geometric mean diameter, number concentration)? If yes, all of them or one specific?
32) Section 2 (lines 131-133): "...all test aerosol was size selected with the help of Differential Mobility Analyzer (DMA) column, part of Scanning Mobility Particle Sizer (SMPS)."
There is no need to mention that the DMA was part of an SMPS system. However, the appropriate reference for the DMA, should be added. Model of the DMA (if commercially available) can optionally be added.
33) Section 2 (line 151; Eq. 1): The resolution is not good (i.e., the equation is "blurry").
34) Section 3.1 (lines 184-186): "Figure 2 shows the counting efficiency measured at ambient pressure (~1030 hPa) with ammonium sulfate particles in the range of 135 nm to 300 nm (mobility diameter)."
- i) Instead of using the term "ambient pressure" would that be better to use the term "sea level" or "ground" pressure?
- ii) Better term would be "electrical mobility diameter" instead of " mobility diameter", since the particles were classified by a DMA.
35) Section 3.4 (lines 312-313): "The left “shoulder” peak in the SMPS size distribution (300-400 nm) was mainly caused by missing multiple charge correction, and not real. "
It is not clear what the authors mean by the term "missing multiple charge correction". Please clarify.
36) Section 3.4 (figure 6). The base 10 from the logarithm in the y-axis label can be omitted (i.e., dN/dlogdp instead of dN/dlog10(Dp). In addition the figure caption reads:
"Figure 6, POPS and PCASP size distribution of 500 nm (electrical mobility size) Arizona test dust (ATD) particles."
While in the figure caption the PCASP is referred, in the legend and in the section main text, the other instrument reported except the SMPS is the UHSAS. Please correct accordingly.
37) Section 3.6 (line 374): "Relative humidity (RH) may significantly affect aerosol physical and optical properties. [39-43] "
Reference 41 " Tauber, C.; Brilke, S.; Wlasits, P. J.; Bauer, P. S.; Koberl, G.; Steiner, G.; Winkler, P. M., Humidity effects on the detection of soluble and insoluble nanoparticles in butanol operated condensation particle counters. Atmos Meas Tech 2019, 12, (7), 3659-3671.
refers to condensation particle counters which use butanol as an operating liquid, so it's not that relevant with the OPS technique.
38) Section 3.6 (lines 380-381): "The test aerosol was fed into the residence chamber, where the level of relative humidity was controlled with the variable flow of saturated air."
Please clarify with what the air was saturated.
39) Section 3.6 (lines 381-382): "Residence time in the chamber was long enough for the aerosol to get fully equilibrated with the air."
Not very clear what the authors mean with the phrase "to get fully equilibrated with the air". Do the authors mean that aerosol has to come in an equilibrium with the humidity? If so, please rephrase accordingly.
40) Section 3.6 (lines 382-383): " As expected, the change in RH did not have a significant effect on the PSL particles, as shown in Fig S8, and we did not observe any degradation in POPS"
Why is expected that increased RH conditions do not have a significant effect on PSL particles? Please provide adequate reference(s).
41) Section 3.6 (lines 384-386) "Ammonium sulfate test aerosol showed the expected hysteresis behavior of deliquescence and efflorescence when relative humidity was allowed to change from 40% to 90% and back to 40%."
Please add adequate reference(s) for supporting the "expected" hygroscopic behavior of the ammonium sulfate aerosol.
42) Section 3.6 (lines 388-391): "Smaller growth factor for the optical instruments could be explained by the change of the aerosol refractive index due to humidification (“diluted with water”); the humidified particles were sized much smaller because they produced smaller scattering signal due to lower refractive index."
- i) It would be better if a more scientific term than "diluted with water" is adopted for discussing the effects of water absorption/adsorption on the optical properties of aerosol particles.
- ii) Please provide adequate reference(s) for the second part of the sentence (i.e., "the humidified particles were sized much smaller because they produced smaller scattering signal due to lower refractive index")
43) Section 3.6 (Figure S9): The caption reads: "Figure S9, When the aerosol flow RH = 40%, the estimated aerosol flow temperature under the ambient temperature and RH. (a) for OLI site condition; (b) for SGP site condition. (based on the data between April and October 2018)"
Please clarify the caption.
For example: " Estimated temperature difference between the ambient temperature and the POPS inlet\optical chamber, in order to achieve dry measuring conditions (i.e., RH=40%) at variable ambient RH levels."
44) Section 3.6 (lines 398-399): "Below 40% RH, the hygroscopic growth is limited and usually attributes less than a few percent of diameter change comparing to the dry conditions."
The authors could rephrase this sentence in order to include all phenomena related to the humidification of aerosols, except the change in their size (e.g., changes in optical properties).
45) Section 4.1 (lines 448-450): " Several factors may contribute to the discrepancy: different distortions to the aerosol size distribution due to differences in sampling conditions (isokinetic for POPS and pseudo-isokinetic for PCAP) or losses in lines (POPS)."
The authors attribute the higher particle counts reported by the POPS in respect to the UCASS at sizes < 200 nm, to particle losses in the lines (i.e., for the POP ,which was connected to the isokinetic inlet of the aircraft). This sounds strange, as most commonly losses in the lines are due to particle diffusion and/or impaction. The first mainly affect small particles, while the latter mainly affects bigger particles when sampling downstream abrupt bends or other obstacles. If diffusional losses in the lines that were used for sampling with the POPS were to blame, then it should register lower counts in the smaller sizes. In contrast if impaction losses were present, then one could expect a reduction in POPS particle counting on the bigger size channels. On the other hand the difference in the design/sampling conditions of the inlets (i.e., isokinetic/pseudo-isokinetic) seems like a more plausible explanation, assuming that the POPS has the same counting efficiency with the PCASP at these sizes.
46) Section 4.2 (lines 457-458): " ... one condensation particle counter (model 3007, TSI, lower 457 cut-off size about 10 nm), and a meteorological radiosonde package (iMet 4-RSB)."
- i) The term "D50" can be used for denoting the size at which the CPC exhibits its 50% detection efficiency, instead of the term "cut-off" size which is commonly applied to inertial particle segregators (e.g., impactors, cyclones).
- ii) Optionally, the upper detectable size of the CPC can be added.
47) Section 4.2 (lines 486-488): " We also observed a high abundance of sulfate particles at both heights but a higher fraction at 610m (China et al., paper under preparation)."
As far as I know, forward referencing to papers, especially those not submitted, is not allowed.
48) Section 4.2 (figure 12c): What is the reason of depicting the chemical composition of sampled particles, when this is not discussed in the manuscript and while the data shown there do not contribute to the results/conclusions of the manuscript? It would be better (if possible) that the authors use the results from the scanning electron microscopy to support the size distribution measured by the POPS during the biomass burning event in figure 12b and in the discussion in section 4.2 (cf. major comment #20).
49) Supplement: "Linear size of a fractal aggregate is traditionally described with a radius of gyration Rg; it was empirically found that in the case of aggregate consisting of a large number of monomers radius of gyration (Rg) can be related to mobility diameter Dm as Dm=1.4Rg. The relation between mass parameter (number of monomers in the aggregate N) and radius of gyration Rg can be described as:"
- i) Please define how the mobility diameter was measured/defined. If the authors mean electrical mobility diameter, then they should verify (and mention it accordingly in the text) that it has the same meaning with mobility diameter in the case of fractal-like/irregularly shaped particles.
- ii) Please provide adequate reference(s) for supporting that the radius of gyration can be related to the mobility diameter as "Dm=1.4Rg" . Consider also, under what conditions holds this equation (e.g., under absolute vacuum?)
iii) Provide also the adequate reference(s) for equation 2 in the supplement, as well as for the constant used.
Certain aspects of the above mentioned calculations (e.g., radius of the monomer) could have been better supported if the magnification in figure S6 (b) would be higher.
50) Supplement figure S11: Why the size distribution of the UHSAS is limited at ca. 150 nm or seems shifted (i.e., starting from 60 nm)? Is the UHSAS capable of measuring such small particles (i.e., ca. 60 nm)?
References:
Biskos, G., Paulsen, D., Russell, L. M., Buseck, P. R., and Martin, S. T.. Prompt Deliquescence and Efflorescence of Aerosol Nanoparticles. Atmos. Chem. Phys., 6(12):4633–4642, 2006.
Gao, R. S.; Telg, H.; McLaughlin, R. J.; Ciciora, S. J.; Watts, L. A.; Richardson, M. S.; Schwarz, J. P.; Perring, A. E.; Thornberry, T. D.; Rollins, A. W.; Markovic, M. Z.; Bates, T. S.; Johnson, J. E.; Fahey, D. W., A light-weight, high-sensitivity particle spectrometer for PM2.5 aerosol measurements. Aerosol Sci Tech 2016, 50, (1), 88-99.
Rader, D.J., McMurry P.H. Application of the tandem differential mobility analyzer to studies of droplet growth or evaporation, J. Aerosol Sci., 17, 771-787, 1986.
Seinfeld, J. H. and Pandis, S.N. Atmospheric Chemistry And Physics, 2nd edition, John Wiley & Sons, inc., Hoboken, New Jersey, 2006.
Stanier, C. O., Khlystov, Y., Chan, R. W., Mandiro, M. and Pandis, S. A Method for the In Situ Measurement of Fine Aerosol Water Content of Ambient Aerosols: The Dry-Ambient Aerosol Size Spectrometer (DAASS), Aerosol Sci Tech 2004, 38, 215-228.
TSI Inc., Specifications sheet, Hand-Held Condensation Particle Counter Model 3007, https://www.tsi.com/getmedia/8c0677b4-6cda-43b6-9a74-1c96acc30d4e/3007_5001117_A4?ext=.pdf, accessed online on 11/10/2020.

Author Response
We sincerely appreciate the comments and suggestions from our reviewer. Thank you very much for helping us to improve our manuscript. We address your comments in the attached file (also in blue).

Round 2
Reviewer 1 Report
I think the paper should be published in its current form.
Author Response
Thank you very much for considering the publication of our manuscript.
Reviewer 2 Report
Review on the Revised Version of the Manuscript (Id: 966473) now entitled: "Performance Assessment of Portable Optical Particle Spectrometer (POPS)"
The manuscript has been majorly improved. The authors responded to all of my comments and took adequate actions, when necessary. Further minor improvements can be done. More specifically:
1) Although the main scope of the manuscript is not related to aerosol hygroscopicity, a hygroscopic growth factor of 1.54 at 90% relative humidity (RH) for ammonium sulfate particles is not consistent with differential mobility techniques that probe aerosol hygroscopic properties (i.e., Hygroscopic Tandem DMA systems), nor the theory. The authors have modified the associated sentence that now reads: " The maximum growth factor (the ratio of the aerosol peak size between the condition under the maximum testing RH and the condition of RH =40%) for POPS and UHSAS (around 1.21) is smaller than the SMPS derived one (1.54), which is consistent with the other growth factor study with the differential mobility technique [48, 50].
In the paper of Gao et al. (2006), which is also cited by the authors, the hygroscopic growth factor of >50 nm ammonium sulfate particles is in the order of 1.7 at 90% RH, which is consistent with the theory and other experimental observations. In the above sentence the authors claim that the 1.54 hygroscopic growth factor (i.e., measured in this study at more or less the same maximum RH) is consistent with other studies related to the hygroscopic behavior of ammonium sulfate particles, while it is obvious that this is not the case. I should clarify in this part, that it is essential to maintain the RH of the sheath of the second DMA in HTDMA systems (i.e., differential mobility technique for probing aerosols hygroscopic growth) at the same level as that of the aerosol in order to avoid water evaporation and particle size changes inside the second DMA. This is commonly achieved by modifying the sheath loop of the second DMA by installing a humidification system for the sheath flow, replacing the hygroscopic filters of the sheath loop, etc. In the setup used in this work, the sheath flow loop of the SMPS system (used for probing the size of the humidified ammonium sulfate particles) was most probably not modified. Therefore I suggested (in my previous review) that the most probable cause for the observed hygroscopic growth factor of 1.54 (i.e., this work), which is different than the expected one of ca. 1.7, was the mismatch between the aerosol (90%) and the sheath (<<90%) RH. The references that I suggested in my previous report, discuss and provide evidence for this artefact (i.e., mismatch between aerosol/sheath RH in both tandem DMA and SMPS systems). The authors should not cite them, if not discussing this artefact. In any case this is not a major issue, since the focus of section 3.6 (i.e., this work) is to demonstrate the effects of the RH in the POPS performance and not to measure accurately the hygroscopic growth factor of ammonium sulfate.
However, for avoiding any conflicts between this work and other specifically oriented in aerosol hygroscopicity, studies, the authors should remove the part of the sentence that claims that the observed, with the SMPS, hygroscopic growth factor of ammonium sulfate particles is consistent with other studies using the differential mobility techniques. I believe, that the main finding of this work is that optical instruments will underestimate the aerosols hygroscopic growth factor, due to changes in the optical properties of the aerosols, as the authors correctly state. So, they can just mention that the optical instruments (POPS and UHSAS), although captured the changes in particle size, related to the deliquescence/efflorescence of ammonium sulfate particles, they underestimated the growth factor of these particles. No further explanation is needed, nor (according to my opinion) a direct comparison with the SMPS, since the hygroscopic properties of ammonium sulfate are well studied both theoretically and experimentally at a wide range of RH conditions. If the authors wish to mention the hygroscopic growth factor obtained with their experimental setup and the SMPS, they can, but without claiming that this observation is consistent with other studies and/or the theory.
2) Section 2, second paragraph: " ... all test aerosol was size selected with the help of Differential Mobility Analyzer (DMA) column, (TSI, model 3082).
Minor correction, the model of the DMA should be TSI 3081, as stated further below in the same section. Model 3082 is the model of the "control unit" used in the newest generation of TSI's SMPS/electrical mobility classification systems.
3) Section 4.2, last paragraph: " The above observations indicated both instruments observed the plumb probably from the near site biomass burning event."
Probably mistyped the word plume?
4) Figure S5, supplement: You may remove the matlab "Data Cursor" indication (i.e., yellowish box with X-Y values) from the figure, as it does not seem to provide any further information.
Author Response
We sincerely appreciate the constructive comments and suggestions from our reviewer. Thank you very much for considering the publication of our manuscript. We address your comments in the attached file (also in blue).
